# Isotopic composition of nitrate and particulate organic matter in a pristine dam-reservoir of western India: Implications for biogeochemical processes

Pratirupa Bardhan, S.W.A. Naqvi, Supriya G. Karapurkar, Damodar M. Shenoy, Siby Kurian, Hema Naik.

CSIR-National Institute of Oceanography, Dona Paula, Goa:403004, India.

*Correspondence to*: P.Bardhan (pratirupabardhan@gmail.com)

## *Abstract:*

Isotopic composition of nitrate ($\delta^{15}N$ and $\delta^{18}O$) and particulate organic matter (POM) ($\delta^{15}N$ and $\delta^{13}C$) were measured in Tillari Reservoir, located at the foothills of the Western Ghats, Maharashtra, western India. The reservoir that is stratified during spring-summer and autumn seasons but gets vertically mixed during the Southwest Monsoon (SWM) and winter is characterized by diverse redox nitrogen transformations in space and time. The $\delta^{15}N$ and $\delta^{18}O$ values of nitrate were low ($\delta^{15}N$ = 2-10‰, $\delta^{18}O$ = 5-8‰) during normoxic conditions but increased gradually (highest $\delta^{15}N$=27‰, $\delta^{18}O$=29‰) when anoxic conditions facilitated denitrification in the hypolimnion during spring-early summer. Once nitrate was fully utilized and sulphidic conditions set in, $NH_4^+$ became the dominant inorganic N species, with $\delta^{15}N$ ranging from 1.3 to 2.6‰. Low $\delta^{15}N$ ($\sim$−5‰) and $\delta^{13}C$ (−37‰ to −32‰) of POM co-occurring with high $NH_4^+$ and $CH_4$ in sulphidic bottom waters were probably the consequence of microbial chemosynthesis. Assimilation of nitrate in the epilimnion was the major controlling process on the N-isotopic composition of POM ($\delta^{15}N$ = 2 - 6 ‰). Episodic low $\delta^{15}N$ values of POM (−2 to 0‰) during early summer coinciding with the absence of nitrate might arise from N-fixation, although further work is required to confirm the hypothesis. $\delta^{13}C$-POM in the photic zone ranged between −29‰ and −27‰ for most parts of the year. The periods of mixing were characterized by uniform $\delta^{15}N$-$NO_3^-$ and $\delta^{18}O$-$NO_3^-$ at all depths. Higher POM ( particulate organic carbon (POC) as well as particulate organic nitrogen (PON)) contents and C/N values with lower $\delta^{13}C$-POM during the SWM point to allochthonous inputs. Overall, this study, the first of its kind in the Indian subcontinent, provides an insight into biogeochemistry of Indian reservoirs, using stable carbon and nitrogen isotopes as a tool, where the monsoons play an important role in controlling vertical mixing and dynamics of carbon and nutrients.

## *1.Introduction:*

Nitrogen is an essential macronutrient the availability of which often limits primary production in aquatic ecosystems. It is a polyvalent element that undergoes redox transformation between the terminal oxidation states of $+5$ and $-3$. These transformations involve isotopic fractionation to varying degrees, and so natural abundance of stable isotopes ($^{15}N$ and $^{14}N$) in various N species provides useful insight into nitrogen cycling besides its sources/sinks in the oceanic (Altabet, 1988; Sigman et al., 2005), coastal (Thunell et al., 2004; Hu et al., 2015) and estuarine (Cifuentes et al, 1988; Savoye et al., 2012) water-bodies and sediments. Studies have also been undertaken in freshwater systems like lakes (Pang and Nriagu, 1977; Chen et al., 2014) and reservoirs (Chen and Jia, 2009; Junet et al., 2009). Some of the best studied freshwater ecosystems in this regard are Lake Lugano at the Swiss-Italian border, Lake Kinneret in Israel and Lake Superior in the USA.

In the eutrophic Lake Lugano, the highly depleted $\delta^{13}C$ and $\delta^{15}N$ of the near-bottom POM established the active presence of methanotrophic bacteria during suboxic conditions (Lehmann et al., 2004). Seasonal changes in nitrogen species were reflected in the isotopic composition of particulate organic matter (POM) and dissolved inorganic nitrogen (DIN) compounds in Lake Kinneret (Hadas et al., 2009). Various processes like nitrification, denitrification and $N_2$-fixation were identified with the help of the N isotopes. In Lake Superior, based on nitrate isotopic studies it was possible to identify the increasing inputs of reduced N to the lake and its subsequent nitrification to be the cause behind a century-long increase in the nitrate inventory of the lake, ruling out atmospheric deposition as the other probable cause (Finlay et al., 2007).

There are a large number of natural freshwater lakes as well as man-made reservoirs in India. In fact, India has the third-highest number of dams (around 4300) in the world, after China and USA. However, these systems have not been well investigated for biogeochemical

cycling.. In the very first study of its kind, Narvenkar et al. (2013) sampled eight dam-
reservoirs spread across India and observed strong thermal stratification during summer in all
reservoirs. Six of these reservoirs were found to experience varying degrees of oxygen
depletion in the hypolimnia, ranging from hypoxia to complete anoxia, in spring-summer.
Anoxia has been found to greatly affect the distribution of nitrogen species in these systems.
In order to gain insights into biogeochemical cycling in these poorly investigated water
bodies, we selected the Tillari Reservoir for detailed studies. These included measurements of
natural abundance of nitrogen and oxygen isotopes in nitrate, and nitrogen and carbon
isotopes in POM. These data, first of their kind generated from any Indian freshwater body,
facilitate an understanding of biogeochemical processes (especially involving nitrogen) that
should be typical of any relatively pristine, tropical, monsoon-affected freshwater body.

## *2.Methods:*

### *2.1 Site Description:*

The Tillari Reservoir is situated in the Dodamarg *taluka* in the Sindhudurg district of
Maharashtra ($15^o76'$N, $74^o12'$E, Fig. 1). Created by damming the Tillari River, the reservoir
has a maximum depth of ~50 m and a storage capacity of 0.45 x $10^9$ m$^3$ (Kurian et al. 2012).
The reservoir is located close to the foothills of the Western Ghats, with the drainage basin
having evergreen forests (C3 plant type) as well as grasslands (C3 or C4 plant types)
(Sukumar et al., 1995). The drainage basin of Tillari has low population density, and so the
river water is not much impacted by human activities such as municipal and industrial
discharges, and agriculture. This is reflected by high water quality (Shenoy et al., manuscript
in preparation). The region receives rainfall averaging around 3000 mm annually, almost
entirely between June and September. The evaporation rate in Tillari Reservoir is not known,
but for other Indian reservoirs the evaporative loss is reported to average around 0.2 m
(Subramanya, 2013) per month. Water from Tillari Reservoir is mainly used for irrigation.
Some watershed characteristics of the Tillari Reservoir have been listed in Supplementary
Table 1.
The Tillari Reservoir is a dimictic water body. Relatively low air temperatures and cool winds
descending from the Western Ghats, located immediately to the east of the reservoir, result in
convective mixing and well oxygenated conditions in winter. The water column gets
thermally stratified in spring and remains so until the strong SWM winds and supply of
relatively cold water homogenize the water column again. The water column gets stratified
after the SWM. Stratification during spring-summer leads to anoxic condition that is most
intense (sulphidic in most years) just before the onset of mixing in June-July. A previous
study (Kurian et al, 2012) showed that the occurrence of sulphidic conditions within the
euphotic zone supports anoxygenic photosynthesis by brown sulphur bacteria in this
reservoir. Methane has been found to accumulate in high concentrations below the
thermocline during this period; however, its emissions to the atmosphere are not very high
(Narvenkar et al., 2013). Direct human impacts on nutrient inventory of the reservoir are
relatively minor, as the basin is located amidst thick forests with low human population
density and minimum agricultural activities.

### *2.2 Sampling and field measurements:*


Sampling was conducted at one station located at the deepest part of the reservoir. Water
samples from pre-fixed depths were collected with 5-litre Niskin samplers attached to nylon
ropes and equipped with reversing thermometers to measure temperature. Subsamples for
dissolved oxygen (DO) and hydrogen sulfide ($H_2S$) were collected carefully avoiding air
exchange. Subsamples for nutrients (nitrate and ammonium) were collected in clean 60-ml
HDPE bottles and frozen immediately. Subsamples for stable isotopic analyses were collected
in 5-litre acid-cleaned plastic carboys and transported to the laboratory within 3-4 hours.

### *2.3 Laboratory analyses:*

Dissolved $O_2$ was estimated by the Winkler method (Grasshoff et al., 1983) with a precision of <1 $\mu$M. $NO_3^-$ and $NH_4^+$ were measured using a SKALAR segmented flow analyzer following standard procedures (Grasshoff et al., 1983) with a precision of <0.1 $\mu$M. Dissolved $H_2S$ concentration was determined colorimetrically (Cline, 1969).

### *2.4 Isotopic analyses :*

Sampling for isotopic analyses of POM commenced in March 2010 and continued on a monthly basis till 2012. From 2012 to 2015 samples were collected on a seasonal basis. Samples for nitrate isotopic measurements were collected from 2011. The facility for nitrate isotope analysis was created in 2014 and samples from 2014 and 2015 were analysed immediately for natural abundance of N and O isotopes. Samples from 2011 and 2012 were also analysed on a selective basis. Samples (upto 3l) for isotopic analyses of POM and DIN (dissolved inorganic nitrogen i.e. $NO_3^-$ and $NH_4^+$) were filtered through precombusted (450° C for 4 hours) 47mm GF/F filters (pore size = 0.7 $\mu$m). The filtrate was used for DIN isotopic measurements and the filter papers were placed in petriplates and frozen immediately.

### *2.4.1 Analyses of $\delta^{15}N$ and $\delta^{18}O$ of $NO_3^-$ :*

Samples for isotopic analysis of nitrate were preserved in two ways. While samples collected in 2011 and 2012 were acidified with HCl to pH 2.5, those taken in 2014 and 2015 were frozen immediately and analysed within a week. Prior to the isotopic analyses, nitrate and nitrite concentrations were measured colorimetrically.  Isotopic analyses of nitrogen and oxygen in $NO_3^-$ were carried out following the "chemical method" (McIlvin and Altabet, 2005) involving reduction of $NO_3^-$ to $NO_2^-$ by cadmium and further reduction to $N_2O$ by

sodium azide in an acetic acid buffer. The resulting $N_2O$ gas in the headspace was purged into
a GasBench II (Thermo Finnigan) and analysed in a Delta V isotope ratio mass spectrometer.
Nitrite concentration was insignificant in most of the samples; sulphamic acid was added in a
few samples that contained nitrite in concentrations exceeding 0.1 µM. Working standards
were prepared in low-nutrient surface seawater (LNSW) collected from the Arabian Sea.
Calibration was done using international nitrate isotope standards USGS-32, USGS-34 and
USGS-35. For further quality assurance, an internal potassium nitrate standard (spanning the
range of nitrate concentration in the samples) was run with each batch of samples.
Magnesium oxide (MgO, Fisher; precombusted for 4 hours at 450˚C) was added to each
sample to raise the pH close to 9 which was followed by addition of cadmium. We used
cadmium powder (Alfa Aesar, −325 mesh, 99.5%) instead of spongy cadmium as mentioned
in McIlvin and Altabet (2005). Each vial was wrapped in aluminium foil and placed on a
horizontal shaker at low speed for 17 hours. After the stipulated time, samples were removed
from the shaker, centrifuged and decanted into clean vials. The nitrite concentrations in the
decanted samples were measured to check the extent of reduction.
Sodium azide (2M solution) and 20% acetic acid were mixed in 1:1 proportion (by volume)
to yield the azide-acetic acid buffer (A-AA buffer) solution. In 20 ml crimp vials, samples
and standards were diluted with LNSW for a final concentration of 20 nmoles and a final
volume of 15 ml. Two international nitrite standards (N23 and N20) were added in this step to
check the efficiency of $N_2O$ production by the buffer. After addition of the A-AA buffer, the
vials were allowed to stand for 1 hour and then the reaction was stopped by adding 0.5ml of
10M NaOH.
The "chemical" method yielded a very low blank (~ 0.5 µM) and worked well for the low
concentration samples. The international standards were run before and after each batch of
samples, while the internal nitrate standards were run after every 5 samples. Analytical
precision (one standard deviation) was better than 0.3‰ for $\delta^{15}N$ and better than 0.7‰ for
$\delta^{18}O$. Results are expressed in $\delta$ notation ($\delta^{15}N$ and $\delta^{18}O$), as per mil (‰) deviation from
atmospheric nitrogen and Vienna Standard Mean Ocean Water (VSMOW), respectively.

### 181 *2.4.2 Analyses of $\delta^{15}N$ of $NH_4^+$:*

Samples for measurements of $\delta^{15}N$- $NH_4^+$ was collected during May 2012 from the anaerobic
hypolimnetic waters.The $\delta^{15}N$ of $NH_4^+$ was measured by the "ammonia diffusion" method
(Holmes et al., 1998). Briefly, 500 ml of sample was collected in duplicates to which 1.5g of
MgO was added to elevate the pH. The diffused $NH_4^+$ was trapped onto acidified glass-fiber
filter sealed between two porous Teflon membranes. The sample bottles were kept in an
incubator-shaker (20˚C, 80 rpm) for two weeks for complete diffusion of $NH_4^+$. After two
weeks, the GF filters were removed from each sample, dried in a $NH_4^+$-free environment,
packed into tin cups and immediately analysed using CF-EA-IRMS. Results were corrected
for blank, percent recovery and fractionation. Analytical precision was better than 0.6‰.

### 191 *2.4.3 Analyses of $\delta^{13}C$ and $\delta^{15}N$ of POM and surface sediment:*

The analyses of $\delta^{13}C$ and $\delta^{15}N$ of POM were usually conducted within 1-2 months of
collection. The frozen filters were acid-fumed with 36% HCl to eliminate carbonates and air
dried in a clean laminar flow. Two aliquots (each of 12 mm diameter) were sub-sectioned
from each filter and packed into tin cups for analysis. Detailed methodology is given in Maya
et al. (2011). The $\delta^{13}C$ and $\delta^{15}N$ of POM along with particulate C and N contents were
analyzed in the same sample using a stable isotope ratio mass spectrometer (Thermo Finnigan
Delta V) connected to an elemental analyser (EURO3000 Eurovector). Results are expressed
as per mil (‰) deviation with respect to PDB (Pee Dee Belemnite) for $\delta^{13}C$ and atmospheric
nitrogen for $\delta^{15}N$. Analytical precision was better than ±0.2‰ as determined from repeated
measurements (after every 5 samples) of a working standard, ε-Amino-n-Caproic Acid
(ACA) having $\delta^{13}C = -25.3‰$ and $\delta^{15}N = 4.6‰$, and a laboratory sediment standard having
$\delta^{13}C = -21‰$ and $\delta^{15}N = 7.5‰$.
Surface sediment collected from the reservoir during the May 2012 field trip was analysed on
only one occasion to investigate its role as an ammonium source. The freeze-dried,
homogenized sample was analyzed following similar protocol.

## 208   *3. Results*

### 209   *3.1 Water column observations*

Based on the vertical temperature distribution it appears that the reservoir gets vertically
mixed through convective overturning in winter (December to February, with the exact
duration of mixing depending upon meteorological conditions prevailing in a given year). In
spring stratification sets in and is the most intense from April to June/July (with a surface-to-
bottom temperature difference of 7-8°C). The water column is again homogenized following
SWM induced mixing and flow of relatively cold water, followed by weaker stratification in
autumn/early winter. A detailed discussion on the physico-chemical parameters is provided in
Shenoy et al. (manuscript under preparation).
The epilimnion was always oxic. During the stratification periods, the DO concentrations
dropped rapidly within the thermocline. The water column became well-oxygenated
following the onset of the southwest monsoon. $H_2S$ was detected below 20 m during the
period of intense stratification (Kurian et al., 2012), with the highest concentration recorded
being 9.88 μM. The occurrence of $H_2S$ was accompanied by the appearance of $CH_4$ and $NH_4^+$.
Upto 160 μM of $CH_4$ and 30 μM of $NH_4^+$ were observed in the anoxic bottom waters during
peak summer (Narvenkar et al., 2013) (Fig. 6).
A thorough analysis of nutrient dynamics in Tillari Reservoir is provided by Naik et al.
(manuscript under preparation). Here we provide a brief description of nitrate profiles during
the study period. Surface water nitrate concentrations were typically low throughout the year
ranging from below detection limit to 0.7 μM. However, the surface nitrate concentrations
were as high as ~10 μM (Fig. 3a) during the SW Monsoon. Nitrate concentrations gradually
increased below the epilimnion during the period of weak stratification. However, with the
depletion of DO, nitrate concentrations in the hypolimnion decreased from 3.6 μM (at 20m)
to 0.3 μM (at 35m), indicating N-loss. Reoxygenation of hypolimnion during the SW
monsoon was accompanied by increase in nitrate concentrations (5-10 μM).

### *3.2 Isotopic composition of nitrate and ammonium*

Large variations in the isotopic composition of nitrate and ammonium were observed in space
and time. Isotopic composition of nitrate in the epilimnion could not be measured on several
occasions due to low concentrations. However, when the measurements could be made it was
observed that the $\delta^{15}N$ and $\delta^{18}O$ values of epilimnetic (0-10 m) $NO_3^-$ were high ($\delta^{15}N$ = 8-
25‰, $\delta^{18}O$ = 24-29‰) (Fig 3b) during the summer stratification presumably due to
autotrophic assimilation whereas relatively lower values ($\delta^{15}N$ = 5-8‰, $\delta^{18}O$ = 12-15‰) were
observed during the monsoon mixing events. Increasing $\delta^{15}N$ and $\delta^{18}O$ of $NO_3^-$, coupled to
decreasing $[NO_3^-]$, were also observed in the suboxic hypolimnion during April and May,
when the water column was strongly stratified. The highest $\delta^{15}N$ values observed were 27.7‰
(in 2014) and 22.4‰ (in 2012) while the corresponding highest $\delta^{18}O$ values were 29.5‰ and
28.8‰, respectively.
The water column remains weakly stratified for a large part of the year, usually from October
to March. A trend of increasing concentrations of isotopically light ($\delta^{15}N$ = 2-8‰ and $\delta^{18}O$ =
5-8‰) nitrate was observed in the hypolimnion along with gradually decreasing levels of
oxygen and ammonium implying the occurrence of nitrification. As the stratification
intensified, this phenomenon was restricted only to the metalimnion. After nitrate was
exhausted, high ammonium build up was observed in the bottom waters. In May 2012, $NH_4^+$
concentrations increased from 0.6 µM at 20m to nearly 12 µM at 40m with a corresponding
decrease in $\delta^{15}N\text{-}NH_4^+$ from 2.6‰ at 20m to 1.3‰ at 40m (Fig. 5a).
Elevated nitrate concentrations occur throughout the water column during the SW monsoon.
The $\delta^{15}N$ and $\delta^{18}O$ of $NO_3^-$ showed little vertical variations at this time. However, interannual
variability was seen in the $\delta^{15}N$ of nitrate (3.94±2.4‰ in 2011, 11.38±1.6‰ in 2014, and
5.47±1.8‰ in 2015), the cause of which will be examined. By contrast, the $\delta^{18}O\text{-}NO_3^-$ values
were relatively less variable (13.01±4.8‰ in 2011, 15.41±2.3‰ in 2014, and 12.46±4.9‰ in

259 2015).

## *3.3 Isotopic and elemental composition of suspended particulate organic matter*

The suspended particulate organic matter in the Tillari Reservoir showed distinct seasonal
and depth-wise variations in its isotopic and elemental compositions (Fig. 2). Primary
productivity in the epilimnion led to higher $\delta^{15}N$ (2‰ to 6‰) and $\delta^{13}C$ (−28‰ to −26‰) in
POM and higher POC (35-60 µM) and PON (4-6 µM) contents as compared to the bottom
water. The molar C/N ratios in the surface waters ranged between 7 and 10. Depleted $\delta^{15}N$
($\sim$−1.4‰) in the epilimnion was observed during the early stratification period (February and
March). As the stratification intensified, the $\delta^{15}N$ and $\delta^{13}C$ of the epilimnetic POM became
heavier, presumably reflecting a gradual enrichment of heavier isotopes in the dissolved
inorganic N and C pools. Both $\delta^{15}N$ and $\delta^{13}C$ decreased with depth with the lowest values
occurring in the anoxic bottom water during peak stratification period. The C/N values in
these waters were in the range of 4-7. In terms of seasonal variability, $\delta^{13}C$ values of POM
were lower during monsoon mixing and became more enriched as the stratification
intensified. The $\delta^{15}N$ values, however, did not depict any distinct seasonal pattern. High POC
(upto 80 µM) and PON (upto 9 µM) along with high C/N (>10) were recorded during the
monsoon season apparently reflecting allochthonous inputs.

## *4. Discussion:*

### *4.1 Epilimnetic processes:*

Nitrate concentrations in surface waters of the Tillari Reservoir varied from below detection
limit during the premonsoon period to 10.7 µM during the SW monsoon. The $\delta^{18}O$ and $\delta^{15}N$
values of nitrate in the epilimnion were high, a signature of assimilation: phytoplankton
prefer nitrate containing $^{14}N$ and $^{16}O$ leaving residual nitrate enriched with $\delta^{15}N$ and $\delta^{18}O$
(Casciotti et al., 2002).  We examined the slopes of the $\delta^{18}O$ vs. $\delta^{15}N$ regression in the surface
water. While a 1:1 line would represent assimilation of epilimnetic nitrate, a steeper slope
would imply assimilation along with the regeneration of nitrate via nitrification (Wankel et
al., 2007). We observed a nearly 1:1 trend for most of the surface water samples during the
summer stratification implying that assimilation exerts the major control on surface $NO_3^-$
isotopic composition (Supplementary Fig. 1).
The isotopic composition of the DIN source exerts the key control on the $\delta^{15}N$ of POM
(Altabet, 2006). The epilimnetic POM in the Tillari Reservoir is expected to have $\delta^{15}N$ less
than or equal to the $\delta^{15}N$-$NO_3^-$. Indeed, the $\delta^{15}N$-POM was always lower than the $\delta^{15}N$ of the
source nitrate (Fig. 3b). The range of $\delta^{13}C$ values of surface-water POM (-32 to -26‰) was
typical of lacustrine autochthonous organic matter (-42 to -24‰, Kendall et al.,2001 and
references therein). As the summer progressed, productivity increased resulting in increased
$CO_2$ uptake and elevated $\delta^{13}C$-POM. During photosynthesis, phytoplankton preferentially

uptake $^{12}C$ leaving the DIC (dissolved inorganic carbon) pool enriched in $^{13}C$. However, when dissolved C is scarce and/or growth rate is high, the phytoplankton would consume the available DIC with reduced or no isotopic discrimination. As the summer progressed at the study location, increased water temperature and low dissolved inorganic nutrient and DIC concentrations would cause the phytoplankton to express reduced isotopic discrimination. This would result in enriched $\delta^{13}C$ of POM. Similar enrichment of $\delta^{13}C$-POM during periods of high productivity have also been observed in other lakes, for e.g., Lake Lugano (Lehmann et al., 2004) and Lake Wauberg (Gu et al., 2006).

In March, when nitrate was close to detection limit, surface $\delta^{15}N$-POM was −1.4‰. The POM resulting from nitrogen fixation by cyanobacteria usually has a $\delta^{15}N$ of 0 to −2‰ (Carpenter et al., 1997). Zeaxanthin, marker pigment of cyanobacteria, was present in significant concentrations (305.1±21 ng l$^{-1}$) within the epilimnion, whereas Chl-*a* concentration was ~1.7 µg l$^{-1}$ (S. Kurian, unpublished data). However, measurements of nitrogen fixation rates in the Tillari Reservoir have yielded very low values during summer (unpublished data). Alternatively, the lower $\delta^{15}N$ values may also result from isotopically light nitrate that is produced in the hypolimnion and diffuses upward into surface waters. Another possible source of isotopically lighter N could be atmospheric deposition, although the magnitude of atmospheric inputs is not expected to be very large during early summer. Further work is required to understand the episodic occurrence of low $\delta^{15}N$-POM.

## *4.2 Biogeochemistry of hypolimnion*

### *4.2.1 Nitrification:*

Stratification in the Tillari Reservoir sets in soon after the decline of the monsoon-fed inflow following which nitrate concentrations increased in oxygenated bottom waters with a concomitant decrease in ammonium concentrations, indicating the occurrence of nitrification.

The nitrate concentrations ranged from below detection limit in the upper 10 m to nearly 10
$\mu M$ close to the bottom. Nitrification occurs in two steps: ammonia oxidation to nitrite
(performed by ammonia oxidising archaea and bacteria) and nitrite oxidation to nitrate
(performed by nitrite oxidising bacteria). Ammonium, the primary N source, undergoes
strong fractionation producing isotopically light nitrate (Delwiche and Stein, 1970, Casciotti
et al., 2003). The $\delta^{15}N$-$NO_3^-$ values ranged from 2-10‰ and the $\delta^{18}O$-$NO_3^-$ ranged from 5-
8‰ during this period. Nitrate accumulation due to atmospheric deposition and microbial
nitrification will have distinct $\delta^{18}O$-$NO_3^-$ values. This is because, while the oxygen atoms in
atmospheric nitrate are derived from interactions between $NO_x$ and $O_3$ in the atmosphere,
those in nitrate produced by nitrification come from dissolved oxygen and water ( Kendall,
1998, Finlay et al., 2007). This is well reflected in the $^{15}N$-$^{18}O$ scatter plot where the $\delta^{18}O$-
$NO_3^-$ data-points from the epilimnion and hypolimnion form completely distinct clusters in
February (Fig 4). As the ammonium pool gets used up, the nitrification rate decreases
accompanied by a decrease in the extent of fractionation (Feigin et al., 1974).
Ammonium, oxygen and carbon dioxide are the major substrates needed for nitrification
(Christofi et al., 1981). While ammonium largely comes from the sediments, oxygen is
supplied from aerated surface waters. During the early stratification period, conducive
conditions exist for nitrifiers to grow within the hypolimnion. However, as the bottom waters
turn increasingly more oxygen-depleted with the intensification of stratification the
"ammonium-oxygen chemocline" (Christofi et al., 1981) moves upward in the water column
and the metalimnion becomes more suitable for the occurrence of nitrification. In April 2014,
$\delta^{18}O$ declined within the thermocline from 34‰ at 5m to 14‰ at 20m owing to nitrification.
Epilimnetic nitrate isotope data are not available for 2012 due to very low nitrate
concentrations. However, the $\delta^{18}O$ declined from 25‰ at 15m to 17‰ at 20m. The $\delta^{15}N$
values in both the years did not show a similar decline, but this is consistent with the results
of several other studies (Böttcher et al., 1990; Burns and Kendall, 2002), where the $\delta^{18}O$ was
found to be better suited for source and process identification than $\delta^{15}N$. It may be noted that
this decoupling of $\delta^{15}N$ and $\delta^{18}O$ was only observed during the peak stratification period at
the thermocline.
The $\delta^{15}N$ and $\delta^{13}C$ values for the POM were generally low during the nitrification period as
also observed in Lake Kinneret (Hadas et al., 2009). The $\delta^{15}N$ varied from −4‰ to 3‰ while
$\delta^{13}C$ varied from −31‰ to −29‰. Assimilation of newly nitrified $NO_3^-$ may be a possible
contributor to POM as indicated by the low $\delta^{15}N$ values.
### *4.2.2 Denitrification:*
During the period of strong stratification, the water column loses oxygen below the
thermocline, which apparently results in N loss. Along with a decrease in nitrate, there also
occurs an increase in $NH_4^+$ concentration. Dissimilatory nitrate reduction is known to be
associated with 1:1 increase in $\delta^{15}N\text{-}NO_3^-$ and $\delta^{18}O\text{-}NO_3^-$ (Granger et al., 2008). Linear
regression of $\delta^{18}O$ versus $\delta^{15}N$ yielded slope values of 0.95 and 0.85 in 2014 and 2012,
respectively. In canonical denitrification, both $\delta^{15}N\text{-}NO_3^-$ and $\delta^{18}O\text{-}NO_3^-$ increase linearly.
The enrichment in isotopic value is ~1 in marine systems (Casciotti et al., 2002, Sigman et
al., 2005, Granger et al., 2008). However, this value is reported to be lower (0.5-0.7) in
freshwater systems (Lehmann et al., 2003 and references therein). The reasons for this
difference are not fully understood. Also, studies in freshwater systems are sparse as
compared to marine systems. In a batch of culture experiments, Granger et al. (2008)
observed that nitrate-reducing enzymes play a role in altering the O to N isotopic enrichment,
with periplasmic dissimilatory nitrate reductase (Nap) expressing a lower enrichment value
(~0.62) than the membrane-bound dissimilatory nitrate reductase. Again, there is a lack of
data on the isotopic expressions of these enzymes at the ecosystem level. Wenk et al. (2014)
attributed the low O:N isotopic effect of ~0.89 to chemolithoautotrophic denitrification,
rather than heterotrophic denitrification, in the northern basin of Lake Lugano.
Our data from the Tillari reservoir indicates the occurrence of denitrification in the suboxic
hypolimnion under stratified conditions. However, this process is restricted to a narrow depth
range of 10-20 m which limits the number of data points. There may be several factors
responsible for the low (<1) isotopic enrichment factor in the Tillari but our data are not
sufficient to identify the exact cause(s).
Assuming the N loss was largely through denitrification, an attempt was made to compute the
fractionation factor using a Rayleigh "closed-system" model (Lehmann et al., 2003).
Although there have been several attempts to compute the nitrogen isotope enrichment
factors in marine systems, ground waters and laboratory cultures (Table 1); similar
information is relatively scarce from freshwater lakes and reservoirs.
The available information on oxygen isotope fractionation is even scarcer. The values of $\varepsilon^{15}$
and $\varepsilon^{18}$ computed by us are $-8.7‰$ and $-10.7‰$, respectively. The $\varepsilon^{15}$ is much lower than
those obtained from laboratory cultures (Olleros, 1983; Table 1) as well as open-ocean OMZs
(Brandes et al., 1998, Voss et al., 2001; Table 1) although it is close to the $\varepsilon^{15}$ reported from
the eutrophic Lake Lugano. Factors controlling denitrification rates in aquatic systems
include temperature, availability of nitrate and organic carbon, oxygen concentration and type
of bacterium involved (Seitzinger et al., 1988, Bottcher et al., 1990, and references therein).
Sedimentary denitrification is known to incur isotope effect ($\varepsilon^{15}$) of ~0‰ due to almost
complete exhaustion of nitrate. The dissolved nitrate concentrations in the Tillari Reservoir
are quite low with the highest values being in the range of 10-12 $\mu M$ (see Results). The
hypolimnetic nitrate concentrations were even lower (< 5 $\mu M$) during periods of anoxia. Low
nitrate availability and sedimentary N-loss may exert major controls on the low $\varepsilon^{15}$ observed
in the Tillari Reservoir.
Denitrification strongly discriminates among the two N isotopes, leaving behind $^{15}$N-enriched
in the residual $NO_3^-$. POM produced by assimilation of this nitrate will also be enriched in
$^{15}$N. However, lower $\delta^{15}$N-PON at these depths implies that $NH_4^+$ was the preferred DIN
source. For instance, observations in April 2012 showed that denitrification was active below
30m and associated with ammonium build-up, there was nearly a 4‰ depletion in $\delta^{15}$N-PON
from 2.5‰ (at 30m) to −2.3‰ (at 40m).

### *4.2.3 Ammonification:*

The isotopic composition of ammonium should reflect that of the sedimentary organic matter
being degraded. In Lake Kinneret (Israel), $\delta^{15}$N-$NH_4^+$ values in the hypolimnion during
stratified conditions ranged from 12 to 17 ‰ reflecting the high $\delta^{15}$N of the sedimentary OM
($\delta^{15}$N = 10‰) (Hadas et al., 2009). In Lake Bled (NW Slovenia), mean $\delta^{15}$N- $NH_4^+$ value of
3.8‰ was similar to that of sedimentary OM ($\delta^{15}$N = 4.5‰) (Bratkic et al., 2012). Likewise,
the sedimentary OM in the Tillari Reservoir had a $\delta^{15}$N of 2.96‰ similar to the $\delta^{15}$N-$NH_4^+$
(1.3-2.6‰) thus establishing remineralization of sedimentary OM as the principal $NH_4^+$
source.
A negative linear relationship between $\delta^{15}$N-PON and ln[$NH_4^+$] was observed (Fig. 5b) which
further indicated uptake of $NH_4^+$. Although this relation was mainly determined by the low
[$NH_4^+$] and high $\delta^{15}$N-PON observed at the top of the hypolimnion (20m), it is important to
include this datapoint to highlight the rapid decline of $\delta^{15}$N-PON over a short depth range.
The fractionation factor ($\varepsilon$) calculated from the slope was −2.4‰. The fractionation factor for
ammonium assimilation has been estimated in several field studies (Cifuentes et al., 1988;
Bratkic et al 2012) as well as in lab cultures with different organisms (green algae, marine
bacteria, etc) (Wada & Hattori, 1978, Wada 1980, Hoch et al 1992). However, such studies in
freshwater lakes and reservoirs are scarce. Bratkic et al. (2012) computed fractionation
factors of −0.8‰ and −1.4‰ for mean ammonium concentrations of 4.7 µM and 3.3 µM
respectively in Lake Bled. Hoch et al. (1992) reported fractionation factor for assimilation by
*Vibrio harveyi*, a marine bacterium, to be between −4‰ and −27‰ for ammonium
concentrations ranging from 23 to 180 µM. The fractionation factor is expected to approach
0‰ for decreased concentrations of ammonium. For the low to moderate ammonium
concentrations recorded (maximum ~12 µM in Figure 5) the fractionation factor computed by
us compares well with previously reported values.

### *4.2.4 Sulphate reduction and evidence for chemosynthesis:*

As the summer intensified and oxidized nitrogen was fully utilized, facultative bacteria
apparently began to utilize sulphate as an electron acceptor as indicated by the accumulation
of $H_2S$.Mass dependent fractionation during microbial degradation of organic matter with
sulphate as an electron acceptor would the residual organic matter enriched in $^{13}C$ and $^{15}N$.
However, following the appearance of $H_2S$, both $\delta^{13}C$-POC and $\delta^{15}N$-PON became more
depleted. The $\delta^{15}N$ values varied between −8‰ and −5‰ and $\delta^{13}C$ values ranged from −37‰
to −32‰ between 30 and 40m depths. The accumulation of $H_2S$ was also accompanied by
significant build-up of $CH_4$ (20-150 µM) and $NH_4^+$ (1-20 µM) (Naik et al., manuscript in
prep.). Increases in POC and PON contents were also observed: from 28 µM to 60 µM for
POC and from 4.7 to 8 µM for PON. Bacterial assimilation of ammonium can explain the
isotopically light nitrogen, but utilization of biogenic methane is known to lead to extremely
low $\delta^{13}C$ values (between −65‰ and −50‰; Whiticar et al., 1986). In our study, the most
depleted $\delta^{13}C$-POC value of −37.8‰ was associated with the highest methane concentration
of 156 µM. Interestingly, in a study carried out in the waters of Lake Baikal in Siberia, very
negative $\delta^{13}C$–DIC values (−28.9 to −35.6‰) were inferred to be derived from methane
oxidation while the $\delta^{13}C$-POC values (−31.7 to −33.5‰ ) were typical of lacustrine organic

matter  (Prokopenko and Williams 2005). The authors explained this lack of correlation between the two C pools by a possible time lag between the peak methane oxidation and peak productivity. Low $\delta^{13}$C-POC ($\sim-37‰$) in Lake Kinneret  was attributed to chemosynthetic C fixation using depleted $\delta^{13}$C-DIC derived from methane oxidation (Hadas et al. 2009). It is important to understand the fate of methane in freshwater systems as they are believed to be significant contributors to atmospheric methane emissions (Bastviken et al., 2004).  The POM isotopic data of the Tillari Reservoir provides evidence for intense microbial chemosynthesis using sulphide, ammonia and methane as energy donors.

### *4.3 Monsoon mixing in Tillari Reservoir:*

The reservoir gets vertically mixed during the months of July, August and September due to a combination of lower atmospheric temperature, strong winds and inflow of relatively cold water during the southwest monsoon. Nitrate concentrations are moderately high throughout the water column, although variable from one year to another. The mean water-column nitrate concentration were 7.26±2.8 μM (n = 10) in 2011, 9.29±0.8 μM (n = 10) in 2014, and 8.13±4.7 μM (n = 9) in 2015. The isotopic composition of nitrate also showed inter-annual variability. While the water column was uniformly nitrate-replete in 2014, the epilimnetic (0-5 m) nitrate concentrations in 2011 and 2015 were markedly lower than those at deeper depths (Fig.7), except at two deepest samples in 2015. This may indicate nitrate uptake by phytoplankton. However, considering its high concentration in rainwater, ammonium is expected to compete with nitrate for phytoplankton uptake. Moreover, the $\delta^{15}$N of nitrate in the epilimnion was lower in 2011 and 2015 than in 2014. In fact, elevated values of $\delta^{15}$N-$NO_3^-$ (>8‰) occurred throughout the water column in 2014 when the nitrate concentration was also generally higher as compared to the other two years. To investigate the cause of this variability, water samples from six upstream stations along the Tillari River along with a

rainwater sample at the main station were collected in 2015. The nitrate concentrations
ranged from 1.8 µM at the most upstream station to 9.4 µM close to our main sampling site.
The ranges of $\delta^{15}N$ and $\delta^{18}O$ of $NO_3^-$ at these stations were 0.4-6.8‰ and 11-27‰,
respectively. The rainwater sample had a nitrate content of 13.89 µM (ammonium = 24.4 µM)
and yielded $\delta^{15}N$ and $\delta^{18}O$ values of −2.9‰ and 88.7‰, respectively. Nitrate in wet
deposition is usually characterised by high $\delta^{18}O$ (> 60‰) (Kendall et al., 2007; Thibodeau et
al., 2013) and low $\delta^{15}N$ (−10 to +5 ‰) (Heaton et al., 2004) values. Unfortunately, the
concentration and isotopic composition of these end members (river runoff and atmospheric
deposition) do not explain the data from the Tillari especially from the 2015. Based on the
high concentration of nitrate in rainwater, it is tempting to suggest that it could be an
important source, but the isotopic data show a mismatch.   The $\delta^{13}C$-POC values in the
epilimnion decreased to nearly −30‰ presumably due to a combination of lower primary
productivity and inputs of organic matter through runoff. Even though the latter was not
measured POC derived from land vegetation is expected to be isotopically light.The POM
data show the ingress of a nearly 30m thick parcel of water from the Tillari River into the
reservoir. This ingress is apparent below 5m depth by distinct $\delta^{13}C$ and $\delta^{15}N$ of POM. The
$\delta^{13}C$-POC increases from −30.9‰ (±0.1‰) in the upper 5m to −25.4‰ (±1‰) between 5m
and 40m. Below 40m, the mean $\delta^{13}C$-POC was −26.5‰ (±1.7‰). The mean $\delta^{15}N$ of the
intermediate water parcel was 5.97±2‰, as compared to 5.49±3‰ in the bottom waters and
3.96±2‰ in the upper 5m. The isotopic data correspond well with the ancillary chemical
parameters, in that the water parcel had a distinct thermal signature (cooler by nearly 2˚C). It
also possessed higher levels of nitrate and lower levels of DO and chlorophyll-*a.*
Thus, looking solely at the high nitrate concentrations in the water column, atmospheric wet
deposition may be a major nitrate source to the water column during the monsoon season.
However, this inference is based on a single measurement where the isotopic composition is
also different. Moreover, the river water is also rain-fed and it is not clear why its isotopic
composition is much lower at the most upstream station.  At the same time, the isotopic
composition of POM indicates influence of the upstream waters. Variable inputs from the
atmosphere and by river runoff to the DIN pool probably account for the interannual
variability, but more studies are needed to identify and quantify these contributions in detail.

## *5. Summary and Conclusions:*

Using stable isotopes of nitrate, ammonium and particulate organic matter, we have been able
to identify distinct water column conditions and transformation processes of reactive nitrogen
in the Tillari Reservoir. The reservoir gets vertically mixed during the southwest monsoon
season as well as in winter; the water column remained stratified during other parts of the
year. The most intense stratification occurs during summer just before the monsoon onset.
Relative   importance   of   microbial   processes   such   as   nitrification,   denitrification,
ammonification and sulphate reduction in the water column varied depending on intensity of
stratification and associated DO levels in the hypolimnion. These processes produced unique
isotopic signatures in the dissolved and particulate matter. Our results suggest the occurrence
of microbial chemosynthesis using methane and ammonium as primary C- and N- sources,
producing organic matter in the anoxic bottom waters that is highly depleted in $^{13}$C and $^{15}$N
content. The thermocline in the Tillari Reservoir has been known to harbour photoautotrophic
sulphur bacteria during peak stratification periods (Kurian et al., 2012). We also found strong
signatures of nitrification within this zone during summer stratification. Autochthonous
production was the principal source of organic matter in the epilimnion which was well-
oxygenated at all times, although productivity was significantly lower during the monsoon
period   due   to   light-limited   conditions.   Nitrate   was   the   preferred   DIN   source   in   the
epilimnion. When nitrate loss occurred in the hypolimnion, the preferred DIN species
switched from nitrate to ammonium. Isotopic measurement of precipitation and upstream
river samples during one seasonal sampling provided some insight into sources of nitrogen,
but the observed inter-annual variability could not be explained. Overall, solar intensity,
water depth and redox conditions appear to be the major factors controlling biogeochemical
cycling in this pristine reservoir.
**Acknowledgements**:
We thank the Director, CSIR-NIO for providing necessary support for this work and the
management body of the Tillari reservoir for permission to carry out this study. This research
was carried out as a part of INDIAS IDEA project funded by the Council of Scientific &
Industrial Research (CSIR). The authors wish to thank Mark Altabet and Laura Bristow for
sharing their expertise. We thank Prof. Sugata Hazra and the School of Oceanographic
Studies, Jadavpur University for their support and encouragement. Puja Satardekar is
acknowledged for analyzing the nutrient samples. Sujal Bandodkar (DTP section, CSIR-NIO)
is thanked for her creative inputs. P. Bardhan thanks CSIR for the award of Senior Research
Fellowship. The authors are also grateful to Ms. Maya MV for her initial assistance in
isotopic analyses and to Mr. H. Dalvi, Mr. A. Methar, Mr. Jonathan and Mr. Sumant for their
help during field work. This is NIO Contribution no. XXXX.

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

Table 1: The values of nitrogen ($\varepsilon^{15}$) and oxygen ($\varepsilon^{18}$) isotope effects for denitrification as
reported from some natural systems as well as laboratory cultures.

| Study Area | $\varepsilon^{15}$ (‰) | $\varepsilon^{18}$(‰) | Reference |
|---|---|---|---|
| Cariaco Basin, Venezuela | | | |
| Beijiang River, China | | 8.5 | |
| Boknis Eck, Baltic Sea | | 15.8 | |
| Lake Lugano, Switzerland | | 6.6 | |
| Groundwater | | 18.3 | |
| Denitrifier culture | | 15 | |
| Denitrifier culture | | | |
| Open-ocean OMZs | | | |
| Shallow groundwater aquifer | | 8 | |
| Tillari reservoir, India | 8.73 | 10.74 | **This study** |


**Figure 1: Map of the sampling location (Tillari Reservoir). T1 shows the main sampling**
**location at the deepest point of the reservoir.**

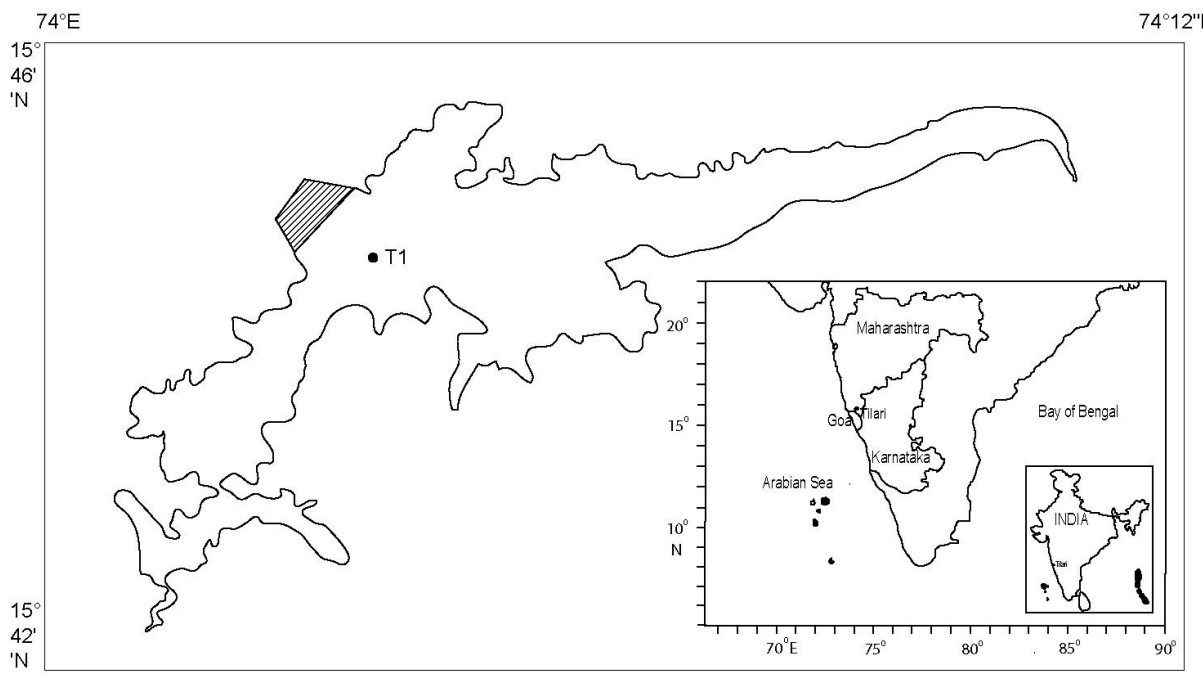
















**Figure 2: Mean annual variations of δ¹⁵N-POM and δ¹³C-POM at the main sampling location.**

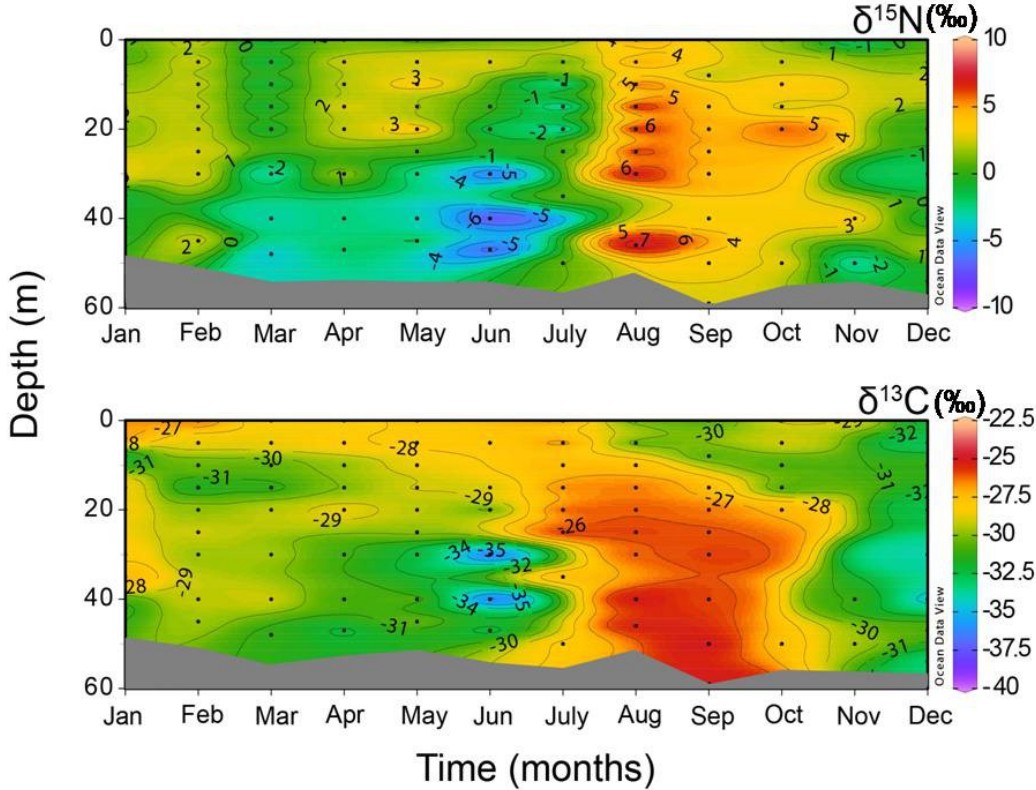

**Figure 3: Time-series of nitrate concentrations (a) and δ¹⁵N of dissolved nitrate and**
**POM in the epilimnion (0-5 m) (b). The isotopic differences between the dissolved and**
**particulate species have been denoted by arrows. Each data point represents one**
**sample. Each data point represents a single sample.**

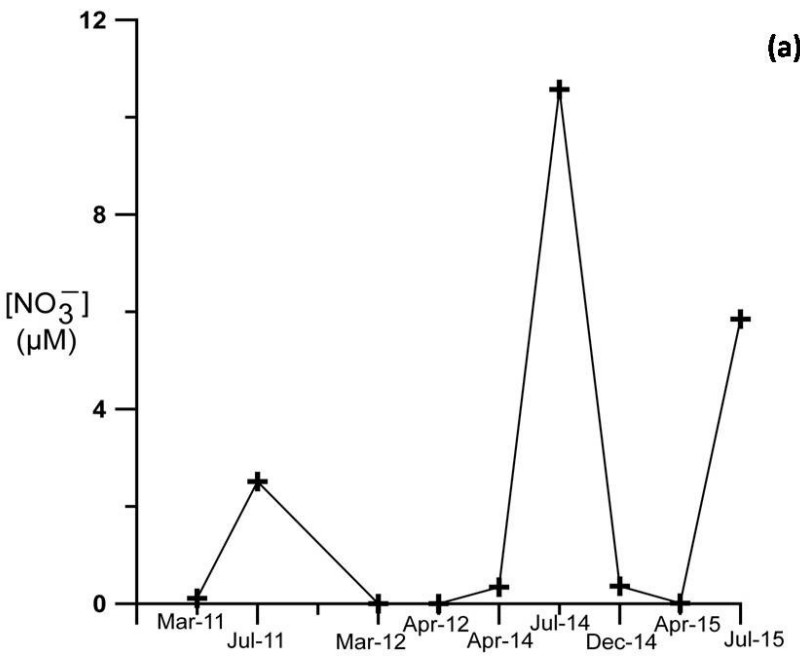


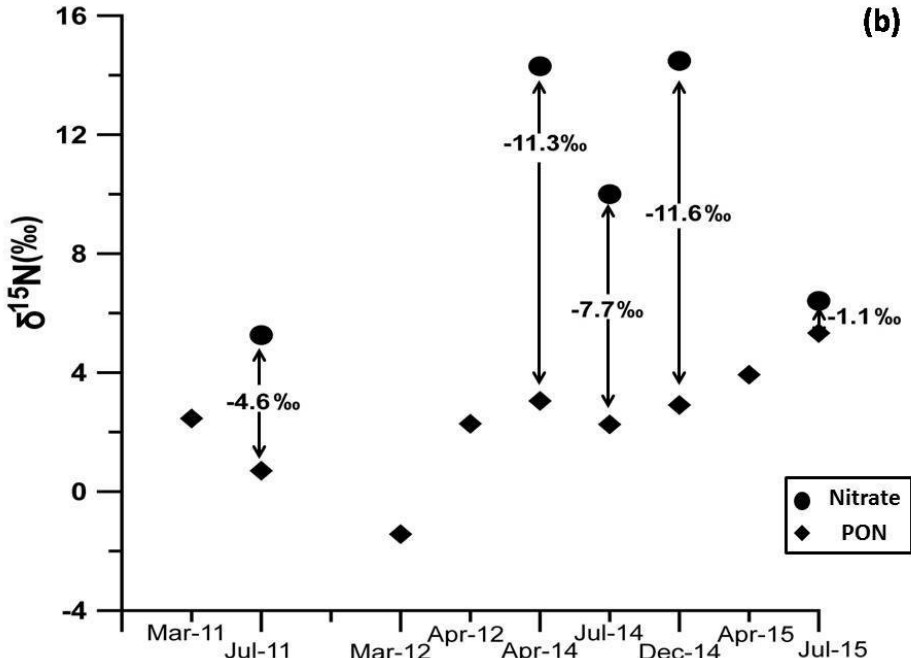


**Figure 4: Nitrogen and oxygen isotopic composition of dissolved nitrate during three**
**different periods in 2014. February represents the early or weak stratification period**
**with two distinct clusters of epilimnetic (0-10 m) and hypolimnetic (15-48 m) samples.**
**April is a period of intense water-column stratification and denitrification signal is**
**observed in the bottom waters. July is a period of monsoon holomixis when the water**
**column has uniformly high nitrate values.**

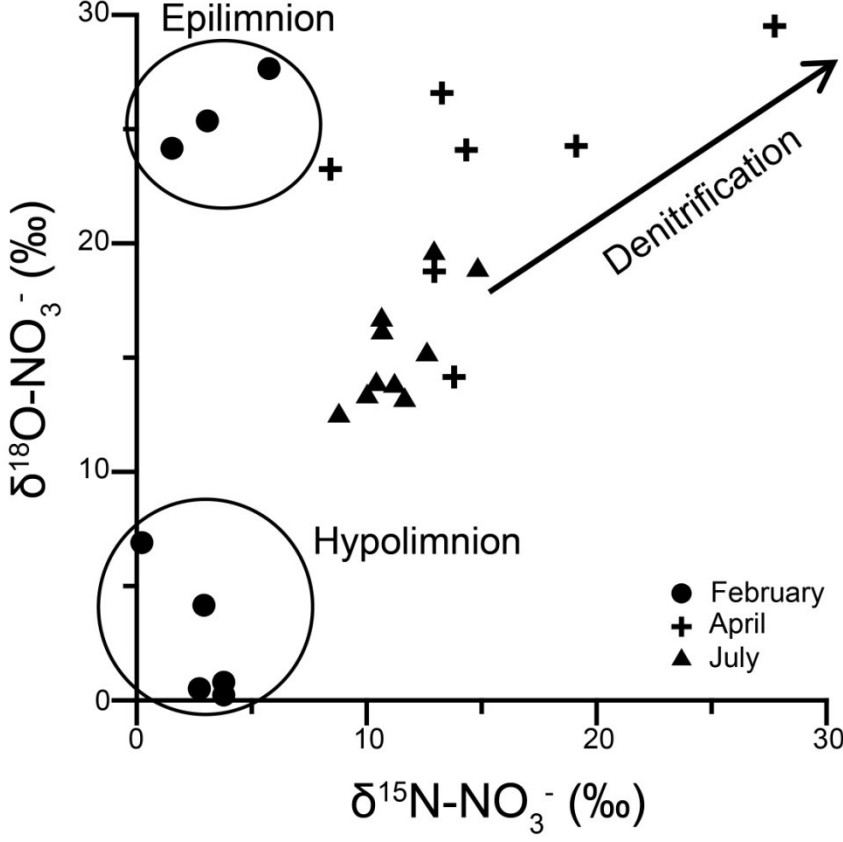








**Figure 5: (a) Depth-wise variations of ammonium concentration and δ¹⁵N-NH₄⁺ in May**
**2012. (b) Plot of δ¹⁵N-PON versus ln(NH₄⁺). The negative linear correlation yields a**
**fractionation factor (ε) of -2.4‰. Each data point represents a single sample.**

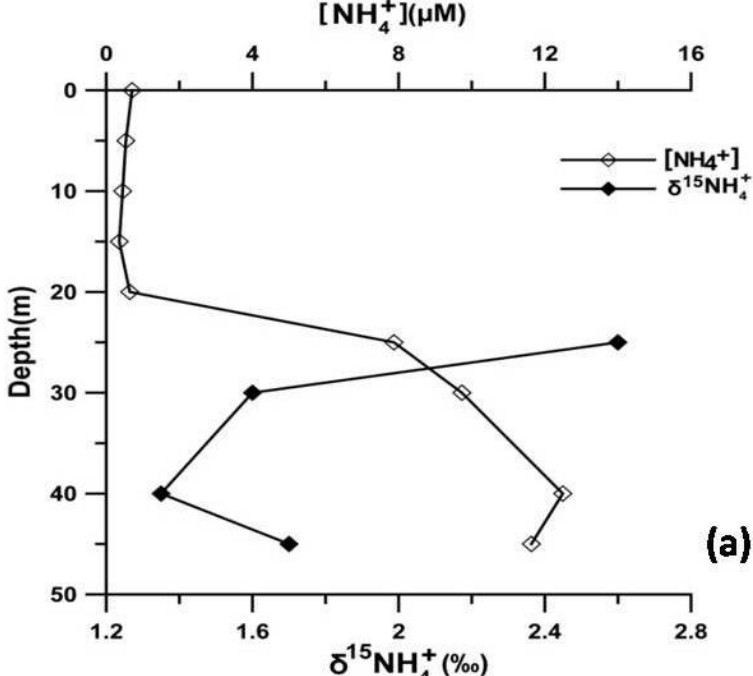


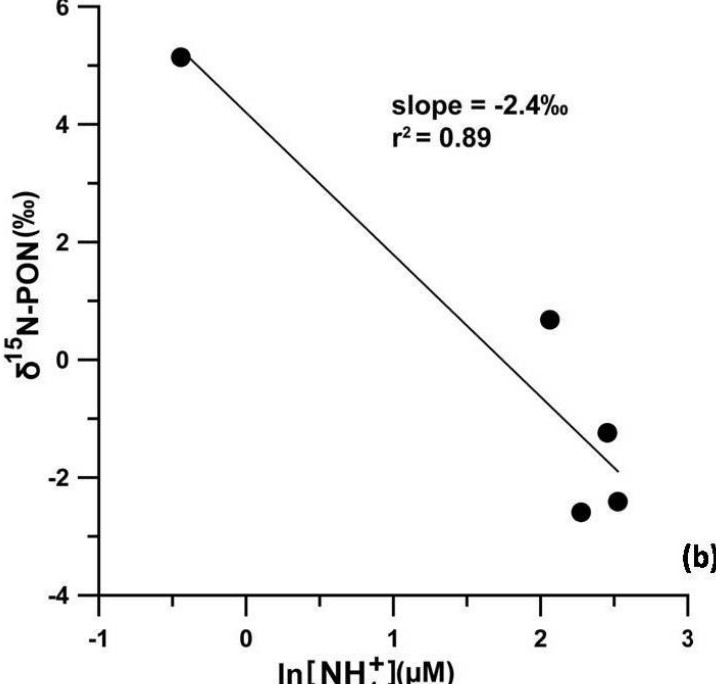


**Figure 6: Schematic diagram depicting major biogeochemical processes taking place in**
**the Tillari Reservoir over an annual cycle. This information is based on monthly**
**sampling in the reservoir for several years (Shenoy et al., manuscript in preparation)**

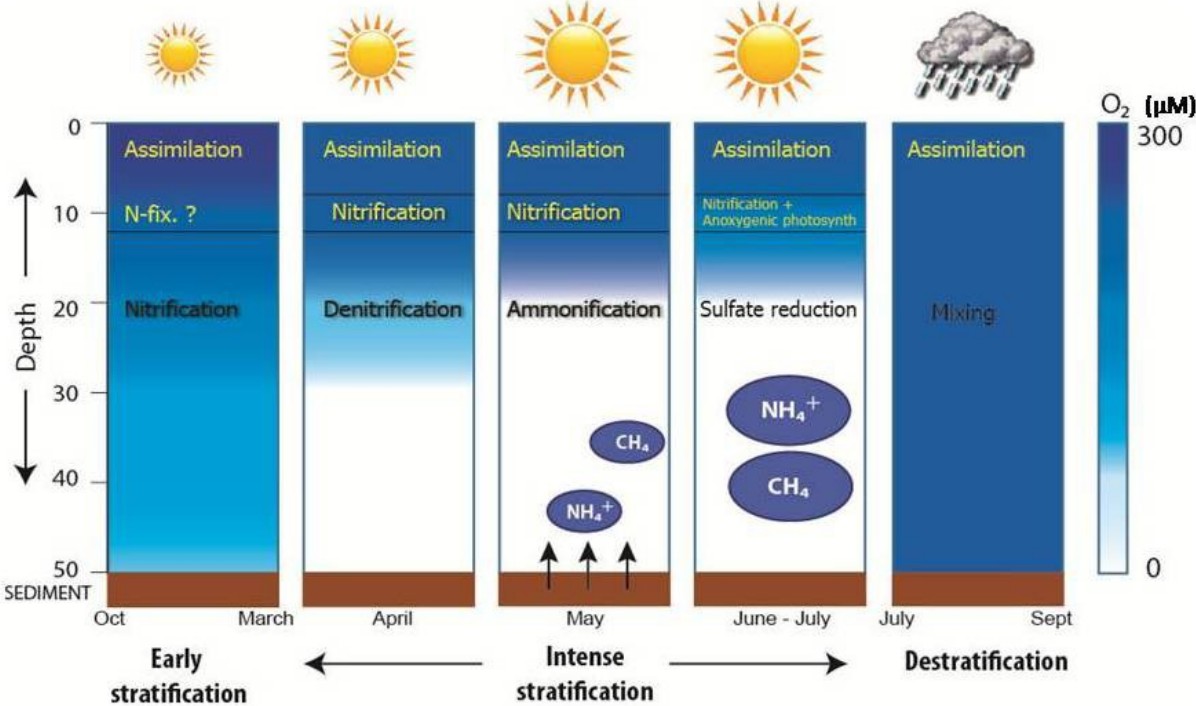













**Figure 7: Vertical profiles of NO$_3^-$ (a) and δ$^{15}$N-NO$_3^-$ (b) during monsoon mixing in 2011,**
**2014 and 2015. Each profile is from one field trip during the peak SWM in a given year**
**with each data point representing one sample.**



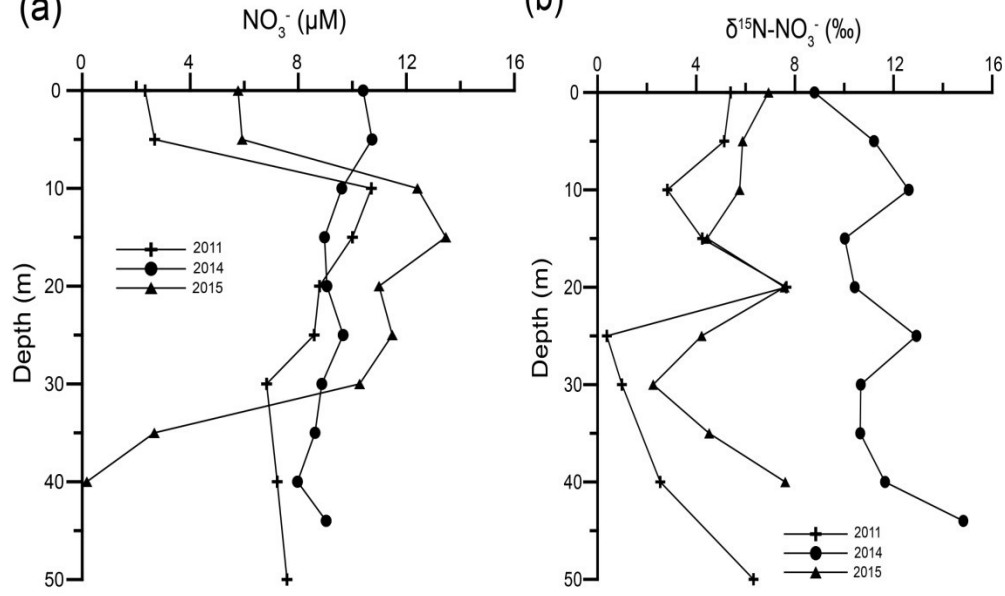
