# Peer review of "Isotopic composition of nitrate and particulate organic matter in a pristine dam-reservoir of western India: Implications for biogeochemical processes"

_Biogeosciences, 2016_

## Referee Comment (RC1) · Anonymous Referee #1 · 17 Aug 2016

This paper reports some interesting results which demonstrates the potential of stable carbon and nitrogen isotopes to gain insight into biogeochemistry of Indian reservoirs where the monsoons play an important role in controlling vertical mixing and dynamics of carbon and nutrients. However, the quality of the text is not sufficient and the data interpretation needs improvements. There are some points that the author gave unclear information, which is showing as follows: 1- Abstract: Please explain the " POC", " PON", " SPOM" and " DIN". 2- Introduction: It is not clear that why did the authors carry out the study? What is the current research progress? 3- Site Description: The description of the study area was not clearly mentioned in this section, such as, land

use, evaporation, water quality. 4- Sampling and field measurements: Please show the distribution of 51 samples in Figure (horizontal and longitudinal). 5- Figure 2, Figure 4 and Figure 5: The data are only from one sample or are the average values? 6- Figure3: Please show the depth of Epilimnion and Hypolimion. 7- Isotopic and elemental composition of suspended particulate organic matter: The data of $\delta$15N and $\delta$13C should be show in table or figure. 7- 4.2.2 Denitrification, L10: why did you get 0.95 and 0.85? 8- I can't find the data from October, November, January, May, June, August and September. Why do you get the diagram to depict different biogeochemical processes taking place in the Tillari Reservoir over an annual cycle in Figure 5.

---

## Referee Comment (RC2) · Anonymous Referee #2 · 4 Sep 2016

Based on the isotopic technique, the paper identifies the biogeochemistry of Indian reservoir where monsoons play an important role in controlling vertical mixing and dynamic of carbon and nutrients. This is important for a better understanding of nutrient cycle in natural freshwater lakes. However, the results and analyses presented here are crude. Thus, a significant work has to be done to improve the overall quality of the manuscript.

Specific comments: (1) Abstract, Line 22-24: The last sentence puts emphasis on the potential of stable carbon and nitrogen isotopes in the study reservoir. However,

throughout the paper the major purpose appears to be identification of biogeochemical processes of the Tillari Reservoir using carbon and nitrogen isotopes. The major purpose of this paper should be made clearer.

(2) The introduction section lacks sufficient overview of previous researches related to biogeochemistry of carbon and nutrients in the reservoirs.

(3) The site description is inadequate. For examples, different vegetation types (C3 plants vs. C4 plants) have distinct values of carbon isotope, which may have an important influence on the carbon isotope of particulate organic matter of the reservoir.

(4) The sampling section lacks a detailed sampling map, which makes it very hard for readers to understand the spatial variations of parameters.

(5) Sampling and analyses. Overall, the sampling time and frequency are not clear. Also, the analyses time and frequency are not clear. For example, when the surface sediment is collected; when nitrogen isotope of NH4 samples are measured.

(6) Results. For this paper, isotopic variations are critical to identify the biogeochemical processes of carbon and nitrogen. Thus, the related isotopic data are needed to present in tables or figures.

(7) Line 10, Page 8: "We observed a nearly 1:1 trend for......" Please display the related variations in figures. The figures can be uploaded as supplementary information.

(8) Line 15, Page 8: "As the summer progressed, productivity increased resulting in increased CO2 uptake and elevated $\delta$13C-POM". This statement is wrong. Values of $\delta$13C-POM are excepted to get more depleted due to the preferential uptake of 12C.

(9) Line 14, Page 8: what is the range of $\delta$13C-POM for surface-water? What is the typical range of lacustrine autochthonous organic matter?

(10) Line 18 and 19, Page 8: the units of "ng/l" and "$\mu$g l-1" should be uniformly expressed as "ng/l" and "$\mu$g/l", or "ng l-1" and "$\mu$g l-1".

(11) Line 20-22, Page 8: in addition to the reasons mentioned, the lower $\delta$15N-POM values may be related to the atmospheric input, which have a low value of $\delta$15N (-2.9‰ Line 1, Page 13) in the study area.

(12) Line 21, Page 9: According to the authors, the decrease of $\delta$18O is due to nitrification. How could you exclude the vertical variations of atmospheric contributions when considering the plentiful rainfall (3000 mm, Line 19, Page 2) in the study area?

(13) Line 10, Page 10: How is "the slope values of 0.95 and 0.85" obtained?

(14) Line 15, Page 10: How are the values of ÉŻ15 and ÉŻ18 computed? The values of ÉŻ15 and ÉŻ18 presented here are wrong. They should be corrected to be -8.7‰ and -10.7‰ respectively.

(15) I don't think Table 1 and Table 2 are necessary. Table 1 is not even cited in the paper. They can be provided as supplementary information.

(16) It is not clear about the description of the data in the title of Fig. 4.

(17) Line 11-19, Page 11: This paragraph compare the fractionation factor of the Tillari reservoir with previous studies. However, what is the conclusion after the comparison?

(18) Line 20, Page 11: the subtitle is "Sulphate reduction and evidence for chemosynthesis". However, there is not any table or figure about the variations of SO4.

(19) Line 21, Page 11: "microbial degradation of organic matter by sulphate......" what does it mean? Does it mean sulphates act as electron acceptor? Please make it clear.

(20) Line 21, Page 12: How can the nitrate isotopic data prove the nitrate uptake? However, the Fig. 4b indicates the uptake of NH4+. Hence, I am wondering whether there is any competitive uptake between NO3 and NH4.

(21) Line 23, Page 12: Only a precipitation sample was collected. It is not enough. Is there any other research about the nitrate isotope of wet deposition in the nearby area?

(22) Line 4-6, Page 13: Where is the data of POM in the Tillari river? How could the POM data prove the input of Tillari river to the reservoir?

(23) Line 12-13, Page 13: "atmospheric wet deposition seems to be the dominant nitrate source to the water column during the monsoon season". However, the related discussion about the atmospheric inputs is extremely scarce throughout the paper.

(24) For Figs. 3 and 6, it is clearer to change symbols in different shapes.

(25) Throughout the paper, some statements lack the related references and some statements lack original references. For example, related references should be added for Line 7, Page 8 and Line 9, Page 10; the original references should be added for Line 8 and Line 12 on Page 9.

---

## Author Comment (AC1) · 31 Oct 2016

Referee Comments: This paper reports some interesting results which demonstrates the potential of stable carbon and nitrogen isotopes to gain insight into biogeochemistry of Indian reservoirs where the monsoons play an important role in controlling vertical mixing and dynamics of carbon and nutrients. However, the quality of the text is not sufficient and the data interpretation needs improvements. Authors' Response: We thank the anonymous referee for her/his contructive comments that have been taken into consideration while revising the manuscript as described below.

[Figure]

Referee Comments: 1- Abstract: Please explain the " POC", " PON", " SPOM" and " DIN" Authors' Response: The abbreviations POC and PON have been expanded in the abstract. In the revised version SPOM has been replaced by POM that is explained in the first line of the abstract.

Referee Comments: 2- Introduction: It is not clear that why did the authors carry out the study? What is the current research progress? Authors' Response: The purpose of this study has been more clearly stated in the revision. The study was undertaken to gain insights into biogeochemical cycling in Indian freshwater reservoirs from which very little information is available so far. The Tillari Reservoir has been selected for detailed investigation that included measurements of natural abundance of nitrogen and oxygen isotopes in nitrate, and nitrogen and carbon isotopes in POM. These data, first of their kind generated from any Indian freshwater body, facilitate an understanding of biogeochemical processes (especially involving nitrogen) that should be typical of any relatively pristine, tropical, monsoon-affected freshwater body.

Referee Comments: 3- Site Description: The description of the study area was not clearly mentioned in this section, such as, land use, evaporation, water quality. Authors' Response: The information sought by the referee has been added under "Site description".

Referee Comments: 4- Sampling and field measurements: Please show the distribution of 51 samples in Figure (horizontal and longitudinal). Authors' Response: The referee mistook "51 samples" as fifty one samples. We meant 5 litre volume. This has now been clarified.

Referee Comments: 5- Figure 2, Figure 4 and Figure 5: The data are only from one sample or are the average values? Authors' Response: Each data point represents one sample. This has now been clarified in figure captions.

Referee Comments: 6- Figure 3: Please show the depth of Epilimnion and Hypolimion. Authors' Response: Epilimnion : 0 – 10 m; Hypolimnion : 15 – 48m. This information

has been added in the caption of Figure 3.

Referee Comments: 7- Isotopic and elemental composition of suspended particulate organic matter: The data of $\delta15N$ and $\delta13C$ should be shown in table or figure. Authors' Response: A figure (tentatively titled Figure A) has now been included that shows mean annual variations of $\delta15N$ and $\delta13C$ of POM.

Referee Comments: 7- 4.2.2 Denitrification, L10: why did you get 0.95 and 0.85? Authors' Response: In canonical denitrification, both $\delta15N\text{-}NO3-$ and $\delta18O\text{-}NO3-$ increase linearly. The enrichment in isotopic value is ~1 in marine systems (Casciotti et al., 2002, Sigman et al., 2005, Granger et al., 2008). However, this value is reported to be lower (0.5-0.7) in freshwater systems (Lehmann et al., 2003 and references therein). The reasons for this difference are not fully understood. Also, studies in freshwater systems are sparse as compared to marine systems. In a batch of culture experiments, Granger et al. (2008) observed that nitrate-reducing enzymes play a role in altering the O to N isotopic enrichment, with periplasmic dissimilatory nitrate reductase (Nap) expressing a lower enrichment value (~0.62) than the membrane-bound dissimilatory nitrate reductase. Again, there is a lack of data on the isotopic expressions of these enzymes at the ecosystem level. Wenk et al. (2014) attributed the low O:N isotopic effect of ~0.89 to chemolithoautotrophic denitrification, rather than heterotrophic denitrification, in the northern basin of Lake Lugano. Our data from the Tillari reservoir indicates the occurrence of denitrification in the suboxic hypolimnion under stratified conditions. However, this process is restricted to a narrow depth range of 10-20 m which limits the number of data points. There may be several factors responsible for the low (<1) isotopic enrichment factor in the Tillari but our data are not sufficient to identify the exact cause(s). This information is included in the revision.

Referee Comments: 8- I can't find the data from October, November, January, May, June, August and September. Why do you get the diagram to depict different biogeochemical processes taking place in the Tillari Reservoir over an annual cycle in Figure 5. Authors' Response: Figure 5 schematically shows distinct seasonal variations and major biogeochemical processes occurring in the reservoir. This information in based on regular (monthly) monitoring of the reservoir that includes observations in the months mentioned by the referee. However, the isotope data presented here were not collected on the monthly basis. Shenoy et al. (manuscript under prep.) will provide a more detailed account of intra- and interannual variability in the reservoir based on monthly sampling. This has been clarified in the caption of the figure.

References: K. L. Casciotti, D. M. Sigman, M. Galanter Hastings, J. K. Böhlke, and A. Hilkert. Measurement of the oxygen isotopic composition of nitrate in seawater and freshwater using the denitrifier method. Analytical Chemistry, 74:4905–4912, 2002.

J. Granger, D. M. Sigman, M. F. Lehmann, and P. D. Tortell. Nitrogen and oxygen isotope fractionation during dissimilatory nitrate reduction by denitrifying bacteria. Limnology and Oceanography, 53:2533–2545, 2008.

M. F. Lehmann, P. Reichert, S. M. Bernasconi, A. Barbieri, and J. A. McKenzie. Modelling nitrogen and oxygen isotope fractionation during denitrification in a lacustrine redox-transition zone. Geochimica Et Cosmochimica Acta, 67:2529–2542, 2003.

D. M. Sigman, J. Granger, P. J. DiFiore, M. F. Lehmann, R. Ho, G. Cane, and A. van Geen. Coupled nitrogen and oxygen isotope measurements of nitrate along the eastern North Pacific margin. Global Biogeochemical Cycles, 19, 2005.

C. B. Wenk, J. Zopfi, J. Blees, M. Veronesi, H. Niemann, and M. F. Lehmann. Community N and O isotope fractionation by sulfide-dependent denitrification and anammox in a stratified lacustrine water column.Geochimica et Cosmochimica Acta, 125, 551-563, 2014.

Please also note the supplement to this comment:
http://www.biogeosciences-discuss.net/bg-2016-270/bg-2016-270-AC1-supplement.pdf

[Figure]

[Figure]

**Fig. 1.** Figure A: Mean annual variations of δ15N-POM and δ13C-POM at the main sampling location.

---

## Author Comment (AC2) · 31 Oct 2016

Referee Comments: Based on the isotopic technique, the paper identifies the biogeo-chemistry of Indian reservoir where monsoons play an important role in controlling vertical mixing and dynamic of carbon and nutrients. This is important for a better understanding of nutrient cycle in natural freshwater lakes. However, the results and analyses presented here are crude. Thus, a significant work has to be done to improve the overall quality of the manuscript. Authors' Response: We thank the anonymous referee for her/his constructive comments on the manuscript that have enabled us to

improve the quality as detailed below.

Specific comments: Referee Comments: (1) Abstract, Line 22-24: The last sentence puts emphasis on the potential of stable carbon and nitrogen isotopes in the study reservoir. However, throughout the paper the major purpose appears to be identification of biogeochemical processes of the Tillari Reservoir using carbon and nitrogen isotopes. The major purpose of this paper should be made clearer. Authors' Response: We have modified this sentence to "Overall, this study, the first of its kind in the Indian subcontinent, provides an insight into biogeochemistry of Indian reservoirs, using stable carbon and nitrogen isotopes as a tool, where the monsoons play an important role in controlling vertical mixing and dynamics of carbon and nutrients."

Referee Comments: (2) The introduction section lacks sufficient overview of previous researches related to biogeochemistry of carbon and nutrients in the reservoirs. Authors' Response: We have now included a paragraph in the Introduction section that briefly discusses previous researches in some other reservoirs of the world.

Referee Comments: (3) The site description is inadequate. For examples, different vegetation types (C3 plants vs. C4 plants) have distinct values of carbon isotope, which may have an important influence on the carbon isotope of particulate organic matter of the reservoir. Authors' Response: The section on site description has been expanded in response to comments of both referees including vegetation types based on the available information.

Referee Comments: (4) The sampling section lacks a detailed sampling map, which makes it very hard for readers to understand the spatial variations of parameters. Authors' Response: Sampling was carried out only at one location at the deepest part of the reservoir. The station location is now included in the map (Figure 1).

Referee Comments:(5) Sampling and analyses. Overall, the sampling time and frequency are not clear. Also, the analyses time and frequency are not clear. For example, when the surface sediment is collected; when nitrogen isotope of NH4 samples

are measured. Authors' Response: We have tried to improve this section by providing more information. Sampling for isotopic analyses of POM commenced in March 2010 and continued on a monthly basis till 2012. From 2012 to 2015 samples were collected on a seasonal basis. This information is already mentioned in Section 2.4. Samples for nitrate isotopic measurements were collected from 2011. This sentence has been added now. The analyses of $\delta$13C and $\delta$15N of POM were usually conducted within 1-2 months of collection. However, the water samples were filtered within 3-4 hrs of collection and the filter papers were frozen immediately. We have added this sentence in Section 2.4.3. The facility for nitrate isotope analysis was created in 2014 and samples from 2014 and 2015 were analysed immediately for natural abundance of N and O isotopes. Samples from 2011 and 2012 were also analysed on a selective basis. This information was present in Section 2.4. Surface sediment was collected during the May 2012 field trip. We have added this information. Samples for 15N-NH4+ were collected in May 2012. We have added this information in the Methodology Section (Section 2.4.2.). The sample bottles were kept in incubator-shaker immediately on returning to the laboratory following the protocol for the ammonia diffusion method, as mentioned in the Methodology Section. After two weeks' incubation, the measurements were done.

Referee Comments: (6) Results. For this paper, isotopic variations are critical to identify the biogeochemical processes of carbon and nitrogen. Thus, the related isotopic data are needed to present in tables or figures. Authors' Response: A figure (tentatively titled Figure A) has now been included that shows mean annual variations of $\delta$15N and $\delta$13C of POM.

Referee Comments: (7) Line 10, Page 8: "We observed a nearly 1:1 trend for. . .. . ." Please display the related variations in figures. The figures can be uploaded as supplementary information. Authors' Response: We have now added a figure to be included under Supplementary Information (Supplementary Figure 1).

Referee Comments: (8) Line 15, Page 8: "As the summer progressed, productivity increased resulting in increased CO2 uptake and elevated $\delta$13C-POM". This statement

is wrong. Values of $\delta$13C-POM are excepted to get more depleted due to the preferential uptake of 12C. Authors' Response: We argue that higher productivity would result in enrichment of residual DIC with 13C, and the organic matter synthesized would also become increasingly more enriched with this isotope. Of course, the $\delta$13C of POM would still be lower than that of DIC, as stated by the referee, but that is not the point. Similar enrichment of $\delta$13C-POM during periods of high productivity has also been observed in other lakes, for e.g., in Lake Lugano (Lehmann et al., 2004) and in Lake Wauberg (Gu et al., 2006). We have addressed this issue in the revised manuscript.

Referee Comments: (9) Line 14, Page 8: what is the range of $\delta$13C-POM for surface-water? What is the typical range of lacustrine autochthonous organic matter? Authors' Response: The range of $\delta$13C-POM for surface-water is -32 to -26‰ The typical range of lacustrine autochthonous organic matter is -42 to -23‰ (Kendall et al., 2001 and references therein). This has been included in the revision.

Referee Comments: (10) Line 18 and 19, Page 8: the units of "ng/l" and "$\mu$g l-1" should be uniformly expressed as "ng/l" and "$\mu$g/l", or "ng l-1" and "$\mu$g l-1" Authors' Response: Accepted.

Referee Comments: (11) Line 20-22, Page 8: in addition to the reasons mentioned, the lower $\delta$15N-POM values may be related to the atmospheric input, which have a low value of $\delta$15N (-2.9‰ Line 1, Page 13) in the study area. Authors' Response: This possibility has also been included.

Referee Comments: (12) Line 21, Page 9: According to the authors, the decrease of $\delta$18O is due to nitrification. How could you exclude the vertical variations of atmospheric contributions when considering the plentiful rainfall (3000 mm, Line 19, Page 2) in the study area? Authors' Response: The observation referred to in Line 21, Page 9 is from April when the rainfall is negligible and water column is strongly thermally stratified. Dry atmospheric deposition is highly unlikely to cause such a large vertical gradient in $\delta$18O.

Referee Comments: (13) Line 10, Page 10: How is "the slope values of 0.95 and 0.85" obtained? Authors' Response: We considered samples within the suboxic hypolimnion where there was a decrease in nitrate concentration accompanied by an increase in $\delta15N$-NO3$-$ and $\delta18O$-NO3$-$. Linear regression of $\delta18O$ versus $\delta15N$ yielded slope values of 0.95 in 2014 and 0.85 in 2012.

Referee Comments: (14) Line 15, Page 10: How are the values of ÉZ15 and É ËŹ Z18 computed? The values ËŹ of ÉZ15 and É ËŹ Z18 presented here are wrong. They should be corrected to be -8.7‰ ËŹ and -10.7‰ respectively. Authors' Response: We have followed Lehmann et al. (2003) where the calculations for $\varepsilon15$ and $\varepsilon18$ are explained in detail. Using the Rayleigh "closed-system" equation, the $\varepsilon15$ (and $\varepsilon18$) were computed from the slopes of $\delta15N$-NO3$-$ (and $\delta18O$-NO3$-$) versus natural logarithm of f NO3$-$, the fraction of remaining nitrate. The observed maximum [NO3$-$] was considered as the [NO3$-$]initial. f NO3$-$ = [NO3$-$] /[NO3$-$]initial We have corrected the $\varepsilon15$ and $\varepsilon18$ values in the text to -8.7‰ ËŹ and -10.7‰

Referee Comments: (15) I don't think Table 1 and Table 2 are necessary. Table 1 is not even cited in the paper. They can be provided as supplementary information. Authors' Response: We agree that Table 1 can be provided as supplementary information. But we feel Table 2 is important as we are comparing the ÉŻ15 and ÉŻ18 from our study site to those from other systems and also demonstrating the limited data available from freshwater systems in general.

Referee Comments: (16) It is not clear about the description of the data in the title of Fig. 4. Authors' Response: Caption of Fig 4 has been modified as follows: Figure 4: (a) Depth-wise variations of ammonium concentration and $\delta15N$-NH4+ in May 2012. (b) Plot of $\delta15N$-PON versus ln(NH4+). The negative linear correlation yields a fractionation factor (ÉŻ) of -2.4‰

Referee Comments: (17) Line 11-19, Page 11: This paragraph compare the fractionation factor of the Tillari reservoir with previous studies. However, what is the conclusion

after the comparison? Authors' Response: We have added a concluding sentence "For the low to moderate ammonium concentrations recorded (maximum $\sim$12 $\mu$M in Figure 4) the fractionation factor computed by us compares well with previously reported values."

Referee Comments: (18) Line 20, Page 11: the subtitle is "Sulphate reduction and evidence for chemosynthesis". However, there is not any table or figure about the variations of SO4. Authors' Response: H2S is formed during sulphate reduction, and its presence indicates the prevalence of this process. We did not make measurements of sulphate concentration as we thought that these were not needed. In our manuscript, we only discuss this process to interpret the observed variations in $\delta$13C and $\delta$15N of POM during the period H2S accumulates in the water column.

Referee Comments: (19) Line 21, Page 11: "microbial degradation of organic matter by sulphate. . .. . ." what does it mean? Does it mean sulphates act as electron acceptor? Please make it clear. Authors' Response: Yes. We have made appropriate changes in the text.

Referee Comments: (20) Line 21, Page 12: How can the nitrate isotopic data prove the nitrate uptake? However, the Fig. 4b indicates the uptake of NH4+. Hence, I am wondering whether there is any competitive uptake between NO3 and NH4. Authors' Response: Line 21, Page 12 : Accommodating the referee's comments we have thoroughly revised the text.

Referee Comments: (21) Line 23, Page 12: Only a precipitation sample was collected. It is not enough. Is there any other research about the nitrate isotope of wet deposition in the nearby area? Author Comments: We agree that a single sample is not sufficient to reach any meaningful conclusion, and we show that the available data do not explain the observed nitrate concentration and its isotopic composition. In fact, our conclusion is that the composition of end members itself may be highly variable which underlines the need for more measurements. Unfortunately, there are no other data on isotopic

composition of wet deposition in this area.

Referee Comments: (22) Line 4-6, Page 13: Where is the data of POM in the Tillari river? How could the POM data prove the input of Tillari river to the reservoir? Authors' Response: The POM data of the Tillari river was not measured during this study. We hypothesize input of Tillari river by the distinct thermal (colder) signature of the water mass at intermediate depths. As mentioned in the text, this water parcel had higher nitrate, lower DO and chlorophyll-a.

Referee Comments: (23) Line 12-13, Page 13: "atmospheric wet deposition seems to be the dominant nitrate source to the water column during the monsoon season". However, the related discussion about the atmospheric inputs is extremely scarce throughout the paper. Authors' Response: We have modified the text substantially in response to referee's comment.

Referee Comments: (24) For Figs. 3 and 6, it is clearer to change symbols in different shapes. Author Comments: We have modified the figures with different symbols.

Referee Comments: (25) Throughout the paper, some statements lack the related references and some statements lack original references. For example, related references should be added for Line 7, Page 8 and Line 9, Page 10; the original references should be added for Line 8 and Line 12 on Page 9. Author Comments: We have now added the required references. Line 7, Page 8: "The $\delta18O$ and $\delta15N$ values of nitrate in the epilimnion were high, a signature of assimilation: phytoplankton prefer nitrate containing 14N and 16O leaving residual nitrate enriched with $\delta15N$ and $\delta18O$ (Casciotti et al., 2002)" Line 9, Page 10: "Dissimilatory nitrate reduction is known to be associated with 1:1 increase in $\delta15N-NO3-$ and $\delta18O-NO3-$ (Granger et al., 2008)" Line 8, Page 9: "Ammonium, the primary N source, undergoes strong fractionation producing isotopically light nitrate (Delwiche and Stein, 1970, Casciotti et al., 2003)." Line 12, Page9: "This is because, while the oxygen atoms in atmospheric nitrate are derived from interactions between NOx and O3 in the atmosphere, those in nitrate produced by

nitrification come from dissolved oxygen and water ( Kendall, 1998, Finlay et al., 2007)"

Please also note the supplement to this comment:
http://www.biogeosciences-discuss.net/bg-2016-270/bg-2016-270-AC2-supplement.pdf
* * *
[Figure]

[Figure]

**Fig. 1.** Map of the sampling location (Tillari Reservoir). T1 represents the sampling point at the deepest part of the reservoir.

[Figure]

The plot shows a scatter plot with:
- y-axis labeled $\delta^{18}O\text{-}NO_3^-$ (‰) ranging from 24 to 40
- x-axis labeled $\delta^{15}N\text{-}NO_3^-$ (‰) ranging from 0 to 16
- Equation: $y = 0.92x + 22.6$
- $r^2 = 0.97$

**Fig. 2.** Supplementary Figure 1 Scatter plot between $\delta$18O-NO3− vs. $\delta$15N-NO3− of the surface samples during summer. A nearly 1:1 line indicates occurrence of nitrate assimilation.

[Figure]

**Fig. 3.** Figure A: Mean annual variations of $\delta$15N-POM and $\delta$13C-POM at the main sampling location.

[Figure]

Fig. 4. Figure 3(edited)

(a)

NO$_3^-$ (μM)

Fig. 5. Figure 6a (edited)

(b)

δ$^{15}$N-NO$_3^-$ (‰)

Depth (m)

| | | |
|---|---|---|
| + | 2011 |
| ● | 2014 |
| ▲ | 2015 |

**Fig. 6.** Figure 6b (edited)

---

## Author Response (AR1)

**1.COMMENTS FROM AND RESPONSE TO ANONYMOUS REFEREE #1**

**Referee Comments**: This paper reports some interesting results which demonstrates the potential of stable carbon and nitrogen isotopes to gain insight into biogeochemistry of Indian reservoirs where the monsoons play an important role in controlling vertical mixing and dynamics of carbon and nutrients. However, the quality of the text is not sufficient and the data interpretation needs improvements.

Authors' Response: We thank the anonymous referee for her/his contructive comments that have been taken into consideration while revising the manuscript as described below.

**Referee Comments:** 1- Abstract: Please explain the " POC", " PON", " SPOM" and " DIN"

Authors' Response: The abbreviations POC and PON have been expanded in the abstract. In the revised version SPOM has been replaced by POM that is explained in the first line of the abstract.

**Referee Comments**: 2- Introduction: It is not clear that why did the authors carry out the study? What is the current research progress?

Authors' Response: The purpose of this study has been more clearly stated in the revision.
The study was undertaken to gain insights into biogeochemical cycling in Indian freshwater reservoirs from which very little information is available so far. The Tillari Reservoir has been selected for detailed investigation that included measurements of natural abundance of nitrogen and oxygen isotopes in nitrate, and nitrogen and carbon isotopes in POM. These data, first of their kind generated from any Indian freshwater body, facilitate an understanding of biogeochemical processes (especially involving nitrogen) that should be typical of any relatively pristine, tropical, monsoon-affected freshwater body.
.

**Referee Comments:** 3- Site Description: The description of the study area was not clearly mentioned in this section, such as, land use, evaporation, water quality.

Authors' Response: The information sought by the referee has been added under "Site description".

**Referee Comments**: 4- Sampling and field measurements: Please show the distribution of 51 samples in Figure (horizontal and longitudinal).

Authors' Response: The referee mistook "51 samples" as fifty one samples. We meant 5 litre volume. This has now been clarified.

**Referee Comments:** 5- Figure 2, Figure 4 and Figure 5: The data are only from one sample or are the average values?

Authors' Response: Each data point represents one sample. This has now been clarified in figure captions.

**Referee Comments: 6-** Figure 3: Please show the depth of Epilimnion and Hypolimion.

Authors' Response: Figure 3 is numbered as Figure 4 in the revised manuscript. Epilimnion : 0 – 10 m; Hypolimnion : 15 – 48m. This information has been added in the caption of Figure 4.

**Referee Comments**: 7- Isotopic and elemental composition of suspended particulate organic matter: The data of δ15N and δ13C should be shown in table or figure.

Authors' Response: A figure (figure 2) has now been included that shows mean annual variations of $\delta^{15}N$ and $\delta^{13}C$ of POM.

**Referee Comments**:  7- 4.2.2 Denitrification, L10: why did you get 0.95 and 0.85?

Authors' Response: In canonical denitrification, both $\delta^{15}N\text{-}NO_3^-$ and $\delta^{18}O\text{-}NO_3^-$ increase linearly. The enrichment in isotopic value is ~1 in marine systems (Casciotti et al., 2002, Sigman et al., 2005, Granger et al., 2008). However,  this value is reported to be lower (0.5-0.7) in freshwater systems (Lehmann et al., 2003 and references therein). The reasons for this difference are not fully understood. Also, studies in freshwater systems are sparse as compared to marine systems. In a batch of culture experiments, Granger et al. (2008) observed that nitrate-reducing enzymes play a role in altering the O to N isotopic enrichment, with periplasmic dissimilatory nitrate reductase (Nap) expressing a lower enrichment value (~0.62) than the membrane-bound dissimilatory nitrate reductase. Again, there is a lack of data on the isotopic expressions of these enzymes at the ecosystem level. Wenk et al. (2014) attributed the low O:N isotopic effect of ~0.89 to chemolithoautotrophic denitrification, rather than heterotrophic denitrification, in the northern basin of Lake Lugano.
Our data from the Tillari reservoir indicates the occurrence of denitrification in the suboxic hypolimnion under stratified conditions. However, this process is restricted to a narrow depth range of 10-20 m which  limits the number of data points. There may be several factors responsible for the low (<1) isotopic enrichment factor in the Tillari but our data are not sufficient to identify the exact cause(s). This information is included in the revision.

**Referee Comments:** 8- I can't find the data from October, November, January, May, June, August and September. Why do you get the diagram to depict different biogeochemical processes taking place in the Tillari Reservoir over an annual cycle in Figure 5.

Authors' Response: Figure 5 (6 in the revised manuscript) schematically shows distinct seasonal variations and major biogeochemical processes occurring in the reservoir. This information in based on regular (monthly) monitoring of the reservoir  that includes observations in the months mentioned by the referee. However, the isotope data presented here were not collected on the monthly basis. Shenoy et al. (manuscript under prep.) will provide a more detailed account of intra- and interannual variability in the reservoir based on monthly sampling. This has been clarified in the caption of the figure.

**2.COMMENTS FROM AND RESPONSE TO ANONYMOUS REFEREE #2**

**Referee Comments**: Based on the isotopic technique, the paper identifies the biogeochemistry of Indian reservoir where monsoons play an important role in controlling vertical mixing and dynamic of carbon and nutrients. This is important for a better understanding of nutrient cycle in natural freshwater lakes. However, the results and analyses presented here are crude. Thus, a significant work has to be done to improve the overall quality of the manuscript.

Authors' Response: We thank the anonymous referee for her/his constructive comments on the manuscript that have enabled us to improve the quality as detailed below. Specific comments:

**Referee Comments:** (1) Abstract, Line 22-24: The last sentence puts emphasis on the potential of stable carbon and nitrogen isotopes in the study reservoir. However, throughout the paper the major purpose appears to be identification of biogeochemical processes of the Tillari Reservoir using carbon and nitrogen isotopes. The major purpose of this paper should be made clearer.

Authors' Response: We have modified this sentence to "Overall, this study, the first of its kind in the Indian subcontinent, provides an insight into biogeochemistry of Indian reservoirs, using stable carbon and nitrogen isotopes as a tool, where the monsoons play an important role in controlling vertical mixing and dynamics of carbon and nutrients."

**Referee Comments**: (2) The introduction section lacks sufficient overview of previous researches related to biogeochemistry of carbon and nutrients in the reservoirs.

Authors' Response: We have now included a paragraph in the Introduction section that briefly discusses previous researches in some other reservoirs of the world.

**Referee Comments**: (3) The site description is inadequate. For examples, different vegetation types (C3 plants vs. C4 plants) have distinct values of carbon isotope, which may have an important influence on the carbon isotope of particulate organic matter of the reservoir.

Authors' Response: The section on site description has been expanded in response to comments of both referees including vegetation types based on the available information.

**Referee Comments**: (4) The sampling section lacks a detailed sampling map, which makes it very hard for readers to understand the spatial variations of parameters.

Authors' Response: Sampling was carried out only at one location at the deepest part of the
reservoir. The station location is now included in the map (Figure 1).

**Referee Comments**:(5) Sampling and analyses. Overall, the sampling time and frequency are
not clear. Also, the analyses time and frequency are not clear. For example, when the surface
sediment is collected; when nitrogen isotope of NH4 samples are measured.

Authors' Response: We have tried to improve this section by providing more information.
Sampling for isotopic analyses of POM commenced in March 2010 and continued on a
monthly basis till 2012. From 2012 to 2015 samples were collected on a seasonal basis. This
information is already mentioned in Section 2.4. Samples for nitrate isotopic measurements
were collected from 2011. This sentence has been added now.

The analyses of $\delta^{13}C$ and $\delta^{15}N$ of POM were usually conducted within 1-2 months of
collection. However, the water samples were filtered within 3-4 hrs of collection and the filter
papers were frozen immediately. We have added this sentence in Section 2.4.3. The facility
for nitrate isotope analysis was created in 2014 and samples from 2014 and 2015 were
analysed immediately for natural abundance of N and O isotopes. Samples from 2011 and
2012 were also analysed on a selective basis. This information was present in Section 2.4.

Surface sediment was collected during the May 2012 field trip. We have added this
information.

Samples for $^{15}N$-$NH_4^+$ were collected in May 2012. We have added this information in the
Methodology Section (Section 2.4.2.). The sample bottles were kept in incubator-shaker
immediately on returning to the laboratory following the protocol for the ammonia diffusion
method, as mentioned in the Methodology Section. After two weeks' incubation, the
measurements were done.

**Referee Comments:** (6) Results. For this paper, isotopic variations are critical to identify the
biogeochemical processes of carbon and nitrogen. Thus, the related isotopic data are needed
to present in tables or figures.

Authors' Response: A figure (figure 2) has now been included that shows mean annual
variations of $\delta^{15}N$ and $\delta^{13}C$ of POM.

**Referee Comments**: (7) Line 10, Page 8: "We observed a nearly 1:1 trend for. . .. . ." Please
display the related variations in figures. The figures can be uploaded as supplementary
information.

Authors' Response: We have now added a figure to be included under Supplementary
Information (Supplementary Figure 1).

**Referee Comments**: (8) Line 15, Page 8: "As the summer progressed, productivity increased
resulting in increased CO2 uptake and elevated δ13C-POM". This statement is wrong. Values
of δ13C-POM are excepted to get more depleted due to the preferential uptake of 12C.

Authors' Response: We argue that higher productivity would result in enrichment of residual
DIC with $^{13}$C, and the organic matter synthesized would also become increasingly more
enriched with this isotope. Of course, the $\delta^{13}$C of POM would still be lower than that of DIC,
as stated by the referee, but that is not the point. Similar enrichment of $\delta^{13}$C-POM during
periods of high productivity has also been observed in other lakes, for e.g., in Lake Lugano
(Lehmann et al., 2004) and in Lake Wauberg (Gu et al., 2006).

We have addressed this issue in the revised manuscript.

**Referee Comments**: (9) Line 14, Page 8: what is the range of δ13C-POM for surface-water?
What is the typical range of lacustrine autochthonous organic matter?

Authors' Response: The range of $\delta^{13}$C-POM for surface-water is -32 to -26‰. The typical
range of lacustrine autochthonous organic matter is -42 to -23‰ (Kendall et al., 2001 and
references therein). This has been included in the revision.

**Referee Comments**: (10) Line 18 and 19, Page 8: the units of "ng/l" and "μg l-1" should be
uniformly expressed as "ng/l" and "μg/l", or "ng l-1" and "μg l-1"

Authors' Response: Accepted.

**Referee Comments**: (11) Line 20-22, Page 8: in addition to the reasons mentioned, the lower
δ15N-POM values may be related to the atmospheric input, which have a low value of δ15N
(-2.9‰ Line 1, Page 13) in the study area.

Authors' Response: This possibility has also been included.

**Referee Comments:** (12) Line 21, Page 9: According to the authors, the decrease of δ18O is
due to nitrification. How could you exclude the vertical variations of atmospheric
contributions when considering the plentiful rainfall (3000 mm, Line 19, Page 2) in the study
area?

Authors' Response: The observation referred to in Line 21, Page 9 is from April when the
rainfall is negligible and water column is strongly thermally stratified. Dry atmospheric
deposition is highly unlikely to cause such a large vertical gradient in δ18O.

**Referee Comments**: (13) Line 10, Page 10: How is "the slope values of 0.95 and 0.85"
obtained?

Authors' Response: We considered samples within the suboxic hypolimnion where there was
a decrease in nitrate concentration accompanied by an increase in $\delta^{15}$N-NO$_3^-$ and $\delta^{18}$O-NO$_3^-$.
Linear regression of $\delta^{18}$O versus $\delta^{15}$N yielded slope values of 0.95 in 2014 and 0.85 in 2012.

**Referee Comments**: (14) Line 15, Page 10: How are the values of ÉZ15 and É˙Z18
computed? The values ˙ of ÉZ15 and É˙Z18 presented here are wrong. They should be
corrected to be -8.7‰ ˙ and -10.7‰ respectively.

Authors' Response: We have followed Lehmann et al. (2003) where the calculations for $\varepsilon^{15}$

and $\varepsilon^{18}$ are explained in detail. Using the Rayleigh "closed-system" equation, the $\varepsilon^{15}$ (and $\varepsilon^{18}$)

were computed from the slopes of $\delta^{15}N\text{-}NO_3^-$ (and $\delta^{18}O\text{-}NO_3^-$) versus natural logarithm of f

$_{NO3-}$, the fraction of remaining nitrate. The observed maximum $[NO_3^-]$ was considered as the

$[NO_3^-]_{initial}$.

$f_{NO3-} = [NO_3^-] / [NO_3^-]_{initial}$

We have corrected the $\varepsilon^{15}$ and $\varepsilon^{18}$ values in the text to -8.7‰ ˙ and -10.7‰.

**Referee Comments**: (15) I don't think Table 1 and Table 2 are necessary. Table 1 is not even cited in the paper. They can be provided as supplementary information.

Authors' Response: We agree that Table 1 can be provided as supplementary information.

But we feel Table 2 (Table 1 in the revised manuscript) is important as we are comparing the

$\varepsilon^{15}$ and $\varepsilon^{18}$ from our study site to those from other systems and also demonstrating the limited data available from freshwater systems in general.

**Referee Comments**: (16) It is not clear about the description of the data in the title of Fig. 4.

Authors' Response: Caption of Fig 4 has been modified as follows:

**Figure 4: (a) The depth-wise variations of ammonium concentration and $\delta^{15}N\text{-}NH_4^+$ in**

**May 2012. (b) Plot of $\delta^{15}N\text{-}PON$ versus $ln(NH_4^+)$. The negative linear correlation yields**

**a fractionation factor ($\varepsilon$) of -2.4‰.**

**Referee Comments**: (17) Line 11-19, Page 11: This paragraph compare the fractionation factor of the Tillari reservoir with previous studies. However, what is the conclusion after the comparison?

Authors' Response: We have added a concluding sentence "For the low to moderate ammonium concentrations recorded (maximum ~12 μM in Figure 4) the fractionation factor computed by us compares well with previously reported values."

**Referee Comments:** (18) Line 20, Page 11: the subtitle is "Sulphate reduction and evidence for chemosynthesis". However, there is not any table or figure about the variations of SO4.

Authors' Response: $H_2S$ is formed during sulphate reduction, and its presence indicates the prevalence of this process. We did not make measurements of sulphate concentration as we thought that these were not needed. In our manuscript, we only discuss this process to interpret the observed variations in $\delta^{13}C$ and $\delta^{15}N$ of POM during the period $H_2S$ accumulates in the water column.

**Referee Comments:** (19) Line 21, Page 11: "microbial degradation of organic matter by sulphate. . .. . ." what does it mean? Does it mean sulphates act as electron acceptor? Please make it clear.

Authors' Response: Yes. We have made appropriate changes in the text.

**Referee Comments:** (20) Line 21, Page 12: How can the nitrate isotopic data prove the nitrate uptake? However, the Fig. 4b indicates the uptake of NH4+. Hence, I am wondering whether there is any competitive uptake between NO3 and NH4.

Authors' Response: Line 21, Page 12 : Accommodating the referee's comments we have thoroughly revised the text.

**Referee Comments:** (21) Line 23, Page 12: Only a precipitation sample was collected. It is not enough. Is there any other research about the nitrate isotope of wet deposition in the nearby area?

Author Comments: We agree that a single sample is not sufficient to reach any meaningful conclusion, and we show that the available data do not explain the observed nitrate concentration and its isotopic composition. In fact, our conclusion is that the composition of end members itself may be highly variable which underlines the need for more measurements. Unfortunately, there are no other data on isotopic composition of wet deposition in this area.

**Referee Comments**: (22) Line 4-6, Page 13: Where is the data of POM in the Tillari river? How could the POM data prove the input of Tillari river to the reservoir?

Authors' Response: The POM data of the Tillari river was not measured during this study. We hypothesize input of Tillari river by the distinct thermal (colder) signature of the water mass at intermediate depths. As mentioned in the text, this water parcel had higher nitrate, lower DO and chlorophyll-a.

**Referee Comments:** (23) Line 12-13, Page 13: "atmospheric wet deposition seems to be the dominant nitrate source to the water column during the monsoon season". However, the related discussion about the atmospheric inputs is extremely scarce throughout the paper.

Authors' Response: We have modified the text substantially in response to referee's comment.

**Referee Comments**: (24) For Figs. 3 and 6, it is clearer to change symbols in different shapes.

Author Comments: We have modified the figures with different symbols. In the revised manuscript, these are figures 4 and 7.

**Referee Comments:** (25) Throughout the paper, some statements lack the related references and some statements lack original references. For example, related references should be added for Line 7, Page 8 and Line 9, Page 10; the original references should be added for Line 8 and Line 12 on Page 9.

Author Comments: We have now added the required references.

[revised manuscript text omitted]

**Figure 4̶3: Nitrogen and oxygen isotopic composition of dissolved nitrate during three**

**different periods in 2014. February represents the early or weak stratification period**

**with two distinct clusters of epilimnetic (0-10 m) and hypolimnetic (15-48 m) samples.**

**April is a period of intense water-column stratification and denitrification signal is**

**observed in the bottom waters. July is a period of monsoon holomixis when the water**

**column has uniformly high nitrate values.**

[Figure]

[Figure]

**Figure 54: (a) The depth-wise variations of ammonium concentration and δ15N-NH4+ in**

**May 2012. (b)**  **Plot of δ15N-PON versus ln(NH4+).**

**The negative linear**

**correlation yields a fractionation factor (ε) of -2.4‰. Each data point represents a single**

**sample.**

[Figure]

[Figure]

**Figure 65: Schematic diagram depicting major biogeochemical processes taking place in**

**the Tillari Reservoir over an annual cycle. This information is based on monthly**

**sampling in the reservoir for several years (Shenoy et al., manuscript in preparation)**

[Figure]

**Figure 76: Vertical profiles of NO$_3^-$ (a) and $\delta^{15}$N-NO$_3^-$ (b) during monsoon mixing in 2011, 2014 and 2015. Each profile is from one field trip during the peak SWM in a given year with each data point representing one sample.**

[Figure]

---

## Author Response (AR2)

**RESPONSE TO COMMENTS OF THE ASSOCIATE EDITOR**

**Associate Editor comments**: Thank you for replying to the reviewers comments. The revised manuscript is much improved, but I would like you to 1)ensure that Figure 6 is cited in the text and 2)comment on the importance of the single high value to the correlation in 6b. It appears to me that the data point drives the relationship, which should be discussed in the text. The manuscript should be acceptable for publication after addressing that point.

**Author's response:**

We thank the Associate Editor for his inputs.

1)  Figure 6 is cited in the text in the revised version.

2)  We assume the editor is referring to Figure 5b. We have addressed this point in the
 revised manuscript.

**AUTHOR'S CHANGES IN THE MANUSCRIPT**

[revised manuscript text omitted]